# Doubling dolutegravir dosage reduces the viral reservoir in ART-treated people with HIV

Céline Fombellida-Lopez[1,2], Aurelija Valaitienė[2], Lee Winchester[3], Nathalie Maes[4], Patricia Dellot[5], Céline Vanwinge[6], Aurélie Ladang[7], Etienne Cavalier[7], Fabrice Susin[8], Dolores Vaira[8], Marie-Pierre Hayette[8], Catherine Reenaers[9], Michel Moutschen[1,5], Courtney V Fletcher[3], Alexander O Pasternak[2]*[†‡], Gilles Darcis[1,5]*[†‡]

[1]Laboratory of Immunology and Infectious Diseases, GIGA-Institute, University of Liège, Liège, Belgium; [2]Laboratory of Experimental Virology, Department of Medical Microbiology, Amsterdam UMC, University of Amsterdam, Amsterdam, Netherlands; [3]Antiviral Pharmacology Laboratory, University of Nebraska Medical Center, Omaha, United States; [4]Department of Biostatistics and Medico-Economic Information, University Hospital of Liège, Liège, Belgium; [5]Department of General Internal Medicine and Infectious Diseases, University Hospital of Liège, Liège, Belgium; [6]GIGA Flow Cytometry Platform, University of Liège, Liège, Belgium; [7]Department of Clinical Chemistry, University Hospital of Liège, Liège, Belgium; [8]Laboratory of Clinical Microbiology, University Hospital of Liège, Liège, Belgium; [9]Department of Gastroenterology, University Hospital of Liège, Liège, Belgium

**\*For correspondence:**
a.o.pasternak@amsterdamumc.nl (AOP);
gdarcis@chuliege.be (GD)

[†]These authors contributed equally to this work

[‡]Share the last authorship

## eLife Assessment

This **valuable** clinical trial compares the impact of dolutegravir intensification on longitudinal measures of total HIV DNA and day 84 measures of intact HIV DNA. The trial was well-designed, and the paper is easy to read and provides hypothesis generation-level evidence that treatment intensification might decrease intact HIV DNA level in some people after 3 months. The findings are **solid**, with significant limitations being that study endpoints and hypotheses were not precisely defined prior to the trial, and that effect size is limited and inconsistent across trial participants.

**Abstract** Whether antiretroviral therapy (ART) is always completely suppressive, or HIV might continue to replicate at low levels despite ART in some people with HIV (PWH), is still debated. Here, we intensified the ART regimen by doubling dolutegravir (DTG) dosage and investigated the impact of this strategy on HIV blood and tissue reservoirs, immune activation, and inflammation. Twenty HIV-infected adults, who had received a triple ART consisting of 50 mg DTG/600 mg abacavir/300 mg lamivudine pre-intensification and had been suppressed on ART for at least 2 years, were enrolled in a phase 2 randomized clinical trial (https://clinicaltrials.gov/ identifier: NCT05351684). Half of them received an additional 50 mg of DTG for a period of 84 days. As expected, plasma and tissue DTG concentrations significantly increased during the study period in the intensified group but not in the control group. Accordingly, significant decreases in total HIV DNA, intact HIV DNA, and cell-associated unspliced (US) HIV RNA in peripheral blood mononuclear cells, as well as in the US RNA/total DNA ratio, were observed in the intensified group but not in the control group. Intensification also modestly reduced markers of immune activation and exhaustion but had no measurable impact on systemic or tissue inflammation. Together with this, intensification

resulted in a temporary decrease in the CD4/CD8 ratio that returned to baseline by day 84. Our results strongly suggest that the pre-intensification ART regimen was not completely suppressive. If confirmed in larger clinical trials, these results could have an impact on the clinical management of PWH and HIV curative strategies.

## Introduction

Human immunodeficiency virus (HIV) remains a major global public health issue, with an estimated 39.9 million people with HIV (PWH) at the end of 2023. Out of them, 30.7 million (77%) were accessing antiretroviral therapy (ART) (https://www.unaids.org/en/resources/fact-sheet). ART suppresses plasma viral load to undetectable levels, restores immune function, and eliminates the risk of developing AIDS (*Deeks et al., 2013*). Current ART usually consists of one or two nucleotide or nucleoside reverse transcriptase inhibitors and the second or third drug of another class (e.g., a non-nucleoside reverse transcriptase inhibitor, a protease inhibitor (PI), or an integrase strand transfer inhibitor (INSTI)) (https://eacs.sanfordguide.com/). The mechanism of action of ART is interfering with different steps of the viral replication cycle, such as reverse transcription, integration, or viral particle maturation. Importantly, ART only blocks the infection of new cells and does not inhibit HIV gene expression in cells that were infected prior to ART initiation or in the progeny of such cells (*Deeks et al., 2015*).

ART is not curative, and the main barrier to an HIV cure is the presence of latent viral reservoirs that arise due to the ability of HIV to integrate its genome into the host chromosome and persist in a latent form in some of the cells it infects, such as memory CD4+ T cells (*Pasternak and Berkhout, 2023*; *Darcis et al., 2019*). These reservoir cells harbour stably integrated and replication-competent provirus that can reactivate and, after cessation of ART, fuel a rapid viral rebound (*De Scheerder et al., 2019*). HIV reservoirs persist lifelong, slowly decaying over time (*Siliciano et al., 2003*; *Crooks et al., 2015*; *Bachmann et al., 2019*). Several mechanisms have been described to explain the long-term persistence of HIV reservoirs, including the infection of long-lived CD4+ T cells and clonal expansion by homeostatic or antigen-driven cellular proliferation or by HIV integration in or near genes that promote cellular proliferation (*Cohn et al., 2020*; *Coffin and Hughes, 2021*). However, if therapy does not prevent all new infections, then replenishment by de novo virus replication provides an additional potential mechanism of reservoir persistence (*Chun et al., 2005*).

Whether ART is fully suppressive is a topic of considerable and long-standing debate (*Martinez-Picado and Deeks, 2016*). Evidence of residual replication in ART-suppressed PWH has been previously presented (*Sigal et al., 2011*; *Lorenzo-Redondo et al., 2016*; *Pasternak et al., 2012*; *Pasternak et al., 2021*), but other groups published reports that contest this notion (*Bozzi et al., 2019*; *Kearney et al., 2017*; *Van Zyl et al., 2017*; *Kearney et al., 2014*). One strong argument against ongoing viral replication is the absence of detectable virus evolution on ART. However, if residual replication occurs intermittently and at a low level, with short infection chains continuously arising and terminating, then nucleotide substitutions are expected to be sporadic and not linked by temporal structure, unless the sampling is very deep and intensive (*Sigal et al., 2011*; *Conway and Perelson, 2016*). In principle, failure to detect virus evolution does not form an unequivocal proof that residual viral replication does not occur at all: absence of evidence is not evidence of absence. Indeed, Paryad-Zanjani et al. recently developed a mathematical model to track the viral dynamics in lymph nodes (LNs) after the initiation of ART (*Paryad-Zanjani et al., 2023*), which suggested that the methods used by the previous studies to detect genetic divergence were not sufficiently sensitive and that these studies would have been unlikely to detect residual replication.

Over time, this persistent low-level residual replication may contribute to chronic immune activation and inflammation in ART-treated PWH (*Zicari et al., 2019*). Constant stimulation of the immune system could lead to immune exhaustion and premature immune senescence, resulting, over the years, in the development of non-AIDS co-morbidities, such as cardiovascular disease (*Mazzuti et al., 2023*). This replication may occur in anatomical sanctuaries/compartments, notably the LNs and the gut-associated lymphoid tissue, two sites massively infected by HIV (*Chun et al., 2008*). Lower penetration of antiretroviral drugs to these tissues reduces the local drug concentrations and, consequently, their effectiveness in preventing HIV replication (*Fletcher et al., 2014*; *Fletcher et al., 2022*; *Estes et al., 2017*).

One way to investigate the residual replication is to intensify a standard ART regimen and measure the impact of this intensification on latent reservoirs, inflammation, and immune activation. If a reduction of a virological parameter is observed, it would strongly suggest that the ART regimen prior to intensification did not completely prevent virus replication. Several groups have conducted ART intensification trials, either by adding a CCR5 antagonist (maraviroc) or an INSTI (raltegravir or dolutegravir (DTG)) to a standard three-drug regimen. Some of these trials have indeed revealed residual virus replication at baseline, as treatment intensification caused a decrease in immune activation, a decrease in cell-associated HIV RNA in the ileum, and a transient increase in 2-long terminal repeat (2-LTR) circles (*Havlir et al., 2003*; *Ramratnam et al., 2004*; *Buzón et al., 2010*; *Yukl et al., 2010*; *Llibre et al., 2012*; *Vallejo et al., 2012*; *Gutiérrez et al., 2011*; *Hatano et al., 2013*; *Puertas et al., 2018*; *Puertas et al., 2014*). However, other groups have failed to demonstrate any effects of ART intensification on virological parameters (*Dinoso et al., 2009*; *McMahon et al., 2010*; *Gandhi et al., 2012*; *Gandhi et al., 2010*; *Cillo et al., 2015*; *Chaillon et al., 2018*; *Rasmussen et al., 2018*).

In summary, it is still unclear whether ongoing viral replication can occur in the presence of ART. The consequences of residual replication for the HIV curative strategies are important since approaches aimed at eliminating viral reservoirs such as latency reversal (shock-and-kill) (*Deeks, 2012*) are not expected to be fully effective if the virus continues to replicate despite treatment. In the worst-case scenario, the residual replication could lead to significant replenishment of the HIV reservoir upon latency reversal, compensating for any reservoir depletion achieved by this strategy. Therefore, understanding the phenomenon of residual replication and developing strategies to eliminate it are key elements in the quest to cure PWH.

Here, we intensified ART in PWH who had been suppressed on ART for at least 2 years, by doubling the dosage of DTG, an antiretroviral drug that had already been part of the regimen pre-intensification. We investigated the impact of this intensification on HIV blood and tissue latent reservoirs, immune activation, and inflammation.

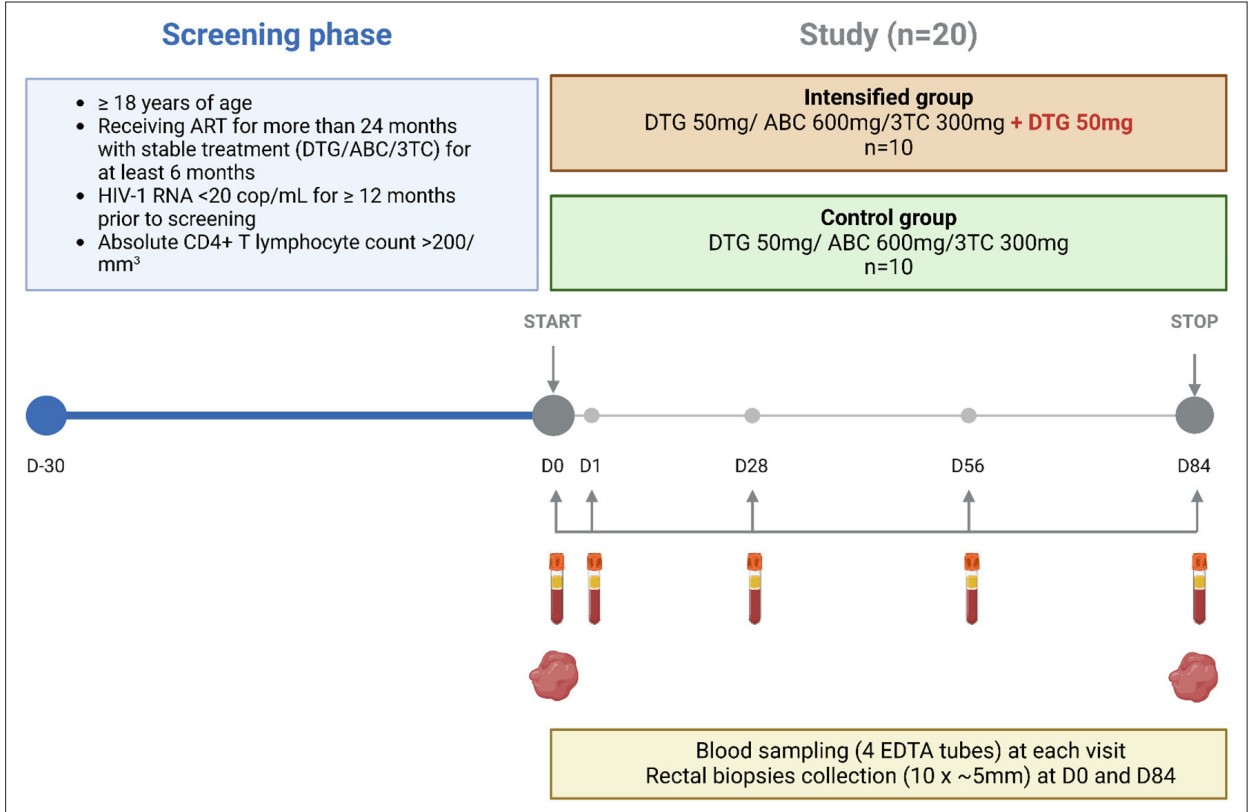

**Figure 1.** Study design. (**A**) Inclusion criteria, study groups, and sampling time points. (**B**) Time points where markers were measured.

## Results

### Study design and participants

Between February and July 2019, twenty ART-suppressed participants receiving a triple antiretroviral regimen consisting of 50 mg DTG, 600 mg abacavir (ABC), and 300 mg lamivudine (3TC) daily were enrolled into two groups as follows: 10 participants continued to receive this regimen (control group) and 10 participants received an additional 50 mg of DTG daily for 84 days (intensified group) as shown in *Figure 1*. Study participants were mostly Caucasian males and the median age was 52 years. The median cumulative time of viral suppression on ART was 7.4 years and participants had maintained continuous viral suppression for a median of 3.9 years before the study. The clinical characteristics of the participants at baseline are shown in *Table 1*.

### ART intensification increases plasma and tissue DTG concentrations

To assess if the desired experimental conditions were achieved, we monitored the concentrations of DTG and 3TC, two antiretroviral drugs that were part of the ART regimen (*Figure 2*, *Figure 2—figure supplement 1*). Baseline plasma and rectal tissue DTG and 3TC concentrations were not significantly different between the intensified and control groups (*Table 1*). At median, baseline DTG concentration in the rectal tissue was 16.7% (interquartile range (IQR), 15.1–24.6%) of the corresponding DTG plasma concentration, strikingly close to the value (17%) reported previously (*Greener et al., 2013*). As expected, plasma and tissue DTG concentrations significantly increased between days 0 and 84 in the intensified group but not in the control group (*Figure 2A, B*). In fact, DTG concentrations almost exactly doubled in plasma (median fold increase, 1.97-fold) and more than doubled in tissue (median fold increase, 2.48-fold) on day 84, as compared to day 0, in the intensified group. However, at day 84, tissue DTG concentration in the intensified group was still much lower (median, 20.0%; IQR, 17.3–22.3%) than the corresponding DTG plasma concentration, reflecting limited penetration of DTG into the tissue. Plasma and tissue 3TC concentrations did not change on day 84 compared to day 0 in both groups, which was expected as the dosage of 3TC was not increased in any group (*Figure 2C, D*). However, we did observe a transient increase in the plasma DTG concentration in the control group and in the plasma 3TC concentration in both groups on days 1 and 56 of the study (*Figure 2—figure supplement 1*), with the concentrations returning to baseline levels by day 84. One possible explanation for this effect could be an increased adherence of the participants to their ART regimens during the study (white coat compliance) (*Podsadecki et al., 2008*).

### ART intensification reduces HIV reservoir

To investigate the effect of treatment intensification on the viral reservoir, we longitudinally measured total HIV DNA and cell-associated unspliced (US) HIV RNA in peripheral blood mononuclear cells (PBMCs) on days 0, 1, 28, 56, and 84 of the study. In addition, we measured intact HIV DNA in PBMCs and total HIV DNA in rectal biopsies on days 0 and 84 of the study. Strikingly, we observed significant longitudinal decreases in total HIV DNA, US RNA, and HIV transcription level per provirus (US RNA/total DNA ratio) in PBMCs in the intensified group (p = 0.0022, p < 0.001, and p < 0.001, respectively) but not in the control group, with significant differences in the relative changes from baseline for two of these three markers between the groups (p < 0.001, p = 0.010, and p = 0.090, respectively) (*Figure 3A–C*, *Figure 3—figure supplements 1–3*). While total HIV DNA in the intensified group transiently decreased in the first 28 days of the study (median (IQR) fold decrease from day 0 to day 28, 2.1 (1.1–4.7)-fold, p = 0.027) and increased afterwards (*Figure 3A*, *Figure 3—figure supplement 1*), US RNA and US RNA/total DNA ratio in the intensified group continued to decrease throughout the whole study period (*Figure 3B, C*, *Figure 3—figure supplements 2 and 3*). By day 84, US RNA and US RNA/total DNA ratio have decreased from day 0 by medians (IQRs) of 5.1 (3.3–6.4)- and 4.6 (3.1–5.3)-fold, respectively (p = 0.016 for both markers). In comparison, in the control group, these markers remained at the baseline level or even increased (*Figure 3A–C*). These effects were confirmed by measurement of intact HIV DNA in PBMCs that was quantified by the intact proviral DNA assay (IPDA) (*Bruner et al., 2019*). We measured a significant decrease in the intact HIV DNA between days 0 and 84 in the intensified group (median (IQR) fold decrease, 2.2 (1.3–3.1)-fold, p = 0.039) but not in the control group (*Figure 3D*). In contrast, total HIV DNA in rectal tissue did not change between days 0 and 84 in both groups (*Figure 3E*). Taken together, these results indicate that the intensification reduced the HIV reservoir in the peripheral blood of ART-treated PWH.

**Table 1.** Clinical characteristics of participants at baseline.

| | All (n = 20) | Control group (n = 10) | DTG group (n = 10) | p* |
|---|---|---|---|---|
| Sex, male | 19 (95.0)† | 9 (90.0) | 10 (100.0) | 1.00 |
| Age, years | 52 (43–60; 25–73) | 52 (40–62; 25–73) | 51 (46–59; 36–66) | 0.94 |
| BMI, kg/m² | 24 (22–26; 18–36) | 25 (22–27; 19–30) | 23 (22–26; 18–36) | 0.62 |
| Ethnicity | | | | |
| Caucasian | 17 (85.0) | 9 (90.0) | 8 (80.0) | 1.00 |
| African | 2 (10.0) | 1 (10.0) | 1 (10.0) | |
| Maghrebi | 1 (5.0) | 0 (0.0) | 1 (10.0) | |
| HLA typing B57/01, negative (n = 19) | 19 (100.0) | 9 (100.0) | 10 (100.0) | - |
| Smoking, smoker, or ex-smoker | 10 (50.0) | 4 (40.0) | 6 (60.0) | 0.66 |
| Time since first positive HIV serology, years | 12.6 (6.8–18.2; 3.5–25.4) | 9.8 (7.3–13.3; 3.5–25.4) | 15.6 (7.6–18.5; 3.5–21.5) | 0.58 |
| Cumulative time of untreated HIV infection, years | 0.9 (0.1–3.5; 0.0–17.2) | 1.8 (0.1–5.0; 0.0–17.2) | 0.6 (0.1–2.1; 0.0–6.1) | 0.47 |
| Cumulative time of viral suppression, years | 7.4 (4.8–9.2; 2.9–16.5) | 5.6 (4.7–7.7; 3.3–16.2) | 7.9 (5.8–12.4; 2.9–16.5) | 0.32 |
| Continuous time of viral suppression before study, years | 3.9 (3.3–5.8; 0.4–10.2) | 4.4 (3.7–6.9; 3.3–10.2) | 3.4 (3.2–4.9; 0.4–7.9) | 0.12 |
| Time on DTG-containing ART regimen before study, years | 3.5 (3.4–3.7; 2.2–3.9) | 3.5 (3.4–3.8; 2.2–3.9) | 3.5 (3.3–3.5; 2.7–3.8) | 0.41 |
| Nadir CD4+ count, cells/mm³ | 277 (147–410; 24–747) | 277 (205–340; 24–521) | 292 (136–508; 30–747) | 0.65 |
| First measured plasma viral load, $\log_{10}$ HIV RNA copies/ml | 4.52 (3.91–5.35; 2.48–6.60) | 4.52 (4.36–4.96; 3.00–6.60) | 4.55 (3.69–5.38; 2.48–5.70) | 0.85 |
| CD4+ count, cells/mm³ | 775 (659–1144; 225–1383) | 732 (672–970; 225–1267) | 886 (668–1217; 492–1383) | 0.39 |
| CD8+ count, cells/mm³ | 745 (637–1288; 377–2017) | 723 (535–931; 377–2017) | 939 (690–1310; 501–1853) | 0.39 |
| CD4/CD8 ratio | 0.94 (0.66–1.47; 0.32–2.24) | 1.00 (0.64–1.36; 0.32–1.82) | 0.93 (0.69–1.56; 0.35–2.24) | 0.82 |
| *Total HIV DNA in PBMCs, copies/10⁶ cells* | 175.5 (70.0–467.3; 15.1–1090) | 99.4 (37.4–231.5; 15.1–333.0) | 421.0 (139.5–764.3; 61.8–1090) | 0.015 |
| *Intact HIV DNA in PBMCs, copies/10⁶ cells* | 55.2 (19.2–87.0; 5.17–624.2) | 29.5 (16.2–97.1; 10.3–624.2) | 58.6 (44.7–144.7; 5.17–292.6) | 0.36 |
| US HIV RNA in PBMCs, copies/μg total RNA | 92.1 (13.3–290.3; 2.53–1580) | 72.2 (12.2–253.8; 6.18–486) | 124.6 (17.0–591.0; 2.53–1580) | 0.58 |
| *US RNA/total DNA ratio in PBMCs* | 0.40 (0.07–1.47; 0.03–3.96) | 1.21 (0.11–2.81; 0.04–3.96) | 0.33 (0.06–0.70; 0.03–2.73) | 0.21 |
| *Total HIV DNA in rectal tissue, copies/10⁶ cells* | 477.0 (271.3–971.5; 7.27–1720) | 547.0 (291.0–1215; 53.4–1720) | 403.0 (194.6–952.5; 7.27–1510) | 0.44 |
| Plasma DTG concentration, ng/mL | 3287 (2602–5087; 237–6593) | 3560 (2953–4455; 1969–5536) | 3266 (2245–5983; 237–6593) | 0.82 |
| Tissue DTG concentration, ng/g | 634 (533–830; 303–1810) | 737 (534–852; 482–1810) | 581 (535–714; 303–1037) | 0.44 |
| Plasma 3TC concentration, ng/ml | 316 (148–731; 50–1616) | 431 (214–941; 102–1616) | 246 (117–439; 50–1156) | 0.26 |

*Table 1 continued on next page*

Table 1 continued

| | All (n = 20) | Control group (n = 10) | DTG group (n = 10) | p* |
|---|---|---|---|---|
| Tissue 3TC concentration, ng/g | 2114 (1417–2345; 90–4495) | 2193 (1590–2784; 1215–4495) | 2114 (1233–2317; 900–3187) | 0.50 |
| CDC classification | | | | - |
| A1 | 4 (20.0) | 1 (10.0) | 3 (30.0) | |
| A2 | 9 (45.0) | 6 (60.0) | 3 (30.0) | |
| A3 | 3 (15.0) | 1 (10.0) | 2 (30.0) | |
| B3 | 1 (5.0) | 0 (0.0) | 1 (10.0) | |
| C3 | 3 (15.0) | 2 (20.0) | 1 (10.0) | |
| HBV status | | | | - |
| Immune | 12 (60.0) | 8 (80.0) | 4 (40.0) | |
| Non-immune, not infected | 5 (25.0) | 2 (20.0) | 3 (30.0) | |
| Isolated HBc Ab | 2 (10.0) | 0 (0.0) | 2 (20.0) | |
| Cured hepatitis B | 1 (5.0) | 0 (0.0) | 1 (10.0) | |
| HCV status | | | | 1.00 |
| Not infected | 19 (95.0) | 10 (10.0) | 9 (90.0) | |
| Recovered | 1 (5.0) | 0 (0.0) | 1 (10.0) | |

*Mann–Whitney tests were used for continuous variables and Fisher's exact tests were used for categorical variables.

†Data are medians (interquartile ranges, followed by ranges) for continuous variables and numbers (percentages) for discrete variables.

The online version of this article includes the following source data for table 1:

**Source data 1.** Numerical data corresponding to *Table 1*.

## ART intensification modestly reduces markers of immune activation and exhaustion

To detect if intensification had an impact on immune activation and exhaustion, we performed longitudinal flow cytometry analysis to assess the expression of cell surface markers of activation (HLA-DR, CD38) and exhaustion (PD-1, TIGIT) on CD4+ and CD8+ T cells (*Figure 4*, *Figure 4—figure supplement 1*). A transient but significant increase in the percentage of CD4+ cells expressing PD-1 (p = 0.022), and a significant decrease in the percentage of CD8+ cells expressing CD38 (p = 0.034) was measured in the control group. A decrease in the percentage of CD4+ cells expressing TIGIT was measured in the intensified group (p = 0.031), with a significant difference compared to the control group (p = 0.048). We also measured a modest yet significant decrease in the percentage of CD8+ cells co-expressing CD38 and HLA-DR in the intensified group (p = 0.049) (*Figure 4*). Overall, these results indicate that intensification modestly reduced markers of immune activation and exhaustion.

## ART intensification does not impact systemic or tissue inflammation

To investigate the effect of intensification on systemic and tissue inflammation, we quantified several inflammatory cytokines in plasma and in rectal tissue. In plasma, we longitudinally measured the levels of IL-1β, IL-6, IFN-γ, IL-17 α, and TNF-α, and in rectal tissue, we measured expression of the same cytokines by quantifying their mRNA levels at days 0 and 84. We observed a significant increase in plasma IL-6 levels in the control group (p = 0.049). Apart from that, no significant change from baseline for any cytokine was observed for both groups in both plasma and tissue (*Figure 5*, *Figure 5—figure supplement 1*). Overall, these results indicate that intensification did not measurably impact systemic or tissue inflammation.

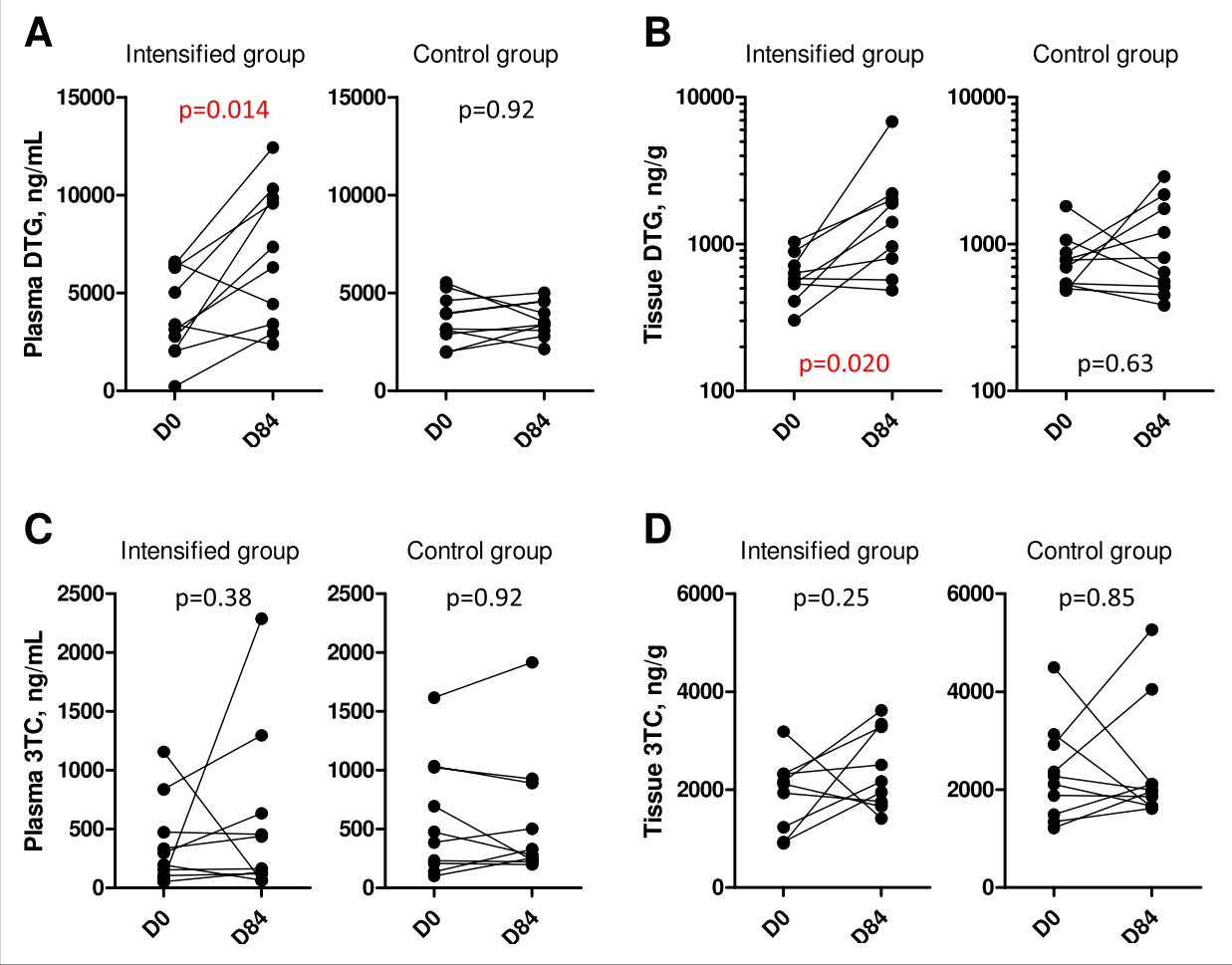

**Figure 2.** Antiretroviral drug concentrations in plasma and rectal tissue. Plasma (**A**) and tissue (**B**) concentrations of DTG, as well as plasma (**C**) and tissue (**D**) concentrations of 3TC, were compared between days 0 (D0) and 84 (D84) in intensified and control groups. Wilcoxon tests were used to calculate statistical significance.

The online version of this article includes the following source data and figure supplement(s) for figure 2:

**Source data 1.** Numerical data corresponding to *Figure 2A*.

**Source data 2.** Numerical data corresponding to *Figure 2B*.

**Source data 3.** Numerical data corresponding to *Figure 2C*.

**Source data 4.** Numerical data corresponding to *Figure 2D*.

**Figure supplement 1.** Plasma concentrations of DTG and 3TC in intensified (blue) and control (red) groups.

**Figure supplement 1—source data 1.** Numerical data corresponding to *Figure 2—figure supplement 1*.

## ART intensification leads to a transient decrease in the CD4/CD8 ratio

We longitudinally measured CD4 and CD8 T-cell counts, CD4/CD8 ratio, as well as plasma levels of soluble CD14 (sCD14), a marker of monocyte/macrophage activation, and of C-reactive protein (CRP), an inflammation marker (*Figure 6*, *Figure 6—figure supplement 1*). No significant changes were observed in the CD4 and CD8 counts in both groups; however, we measured a transient but significant decrease in the CD4/CD8 ratio in the intensified group (p = 0.0081) that returned to the baseline level by day 84 (*Figure 6*). No change in the CD4/CD8 ratio was observed in the control group. We also observed a significant decrease in the sCD14 level in the control group (p < 0.001) and a trend towards an increase in the intensified group (p = 0.079), which resulted in a significant difference between the groups (p < 0.001). Finally, we observed a small but significant increase in the CRP level in the control group (p = 0.045) but not in the intensified group.

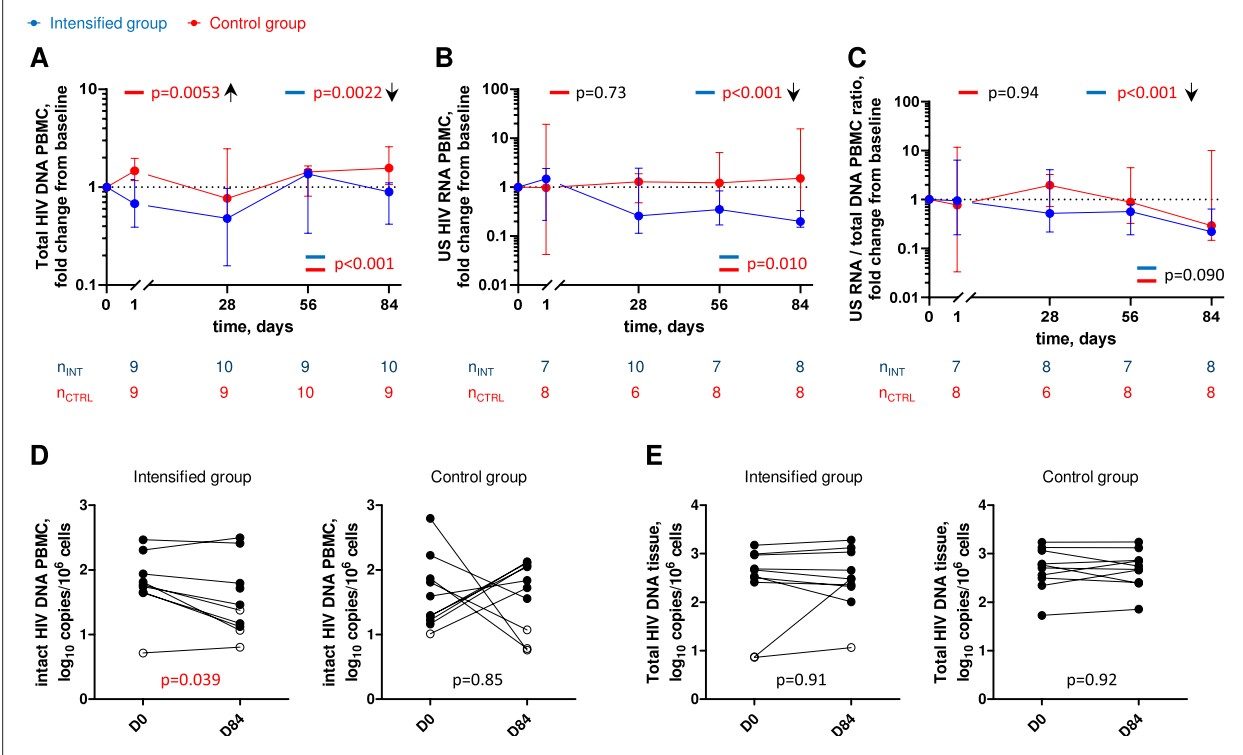

**Figure 3.** Longitudinal dynamics of HIV reservoir markers. Fold change of total HIV DNA in PBMCs (**A**), US HIV RNA in PBMCs (**B**), and US RNA/total DNA ratios (**C**) from baseline on days 1, 28, 56, and 84 of the study in the intensified (blue) and control (red) groups. Median values and interquartile ranges (IQRs) are shown. Linear mixed-effects modelling on $\log_{10}$-transformed values was used to calculate statistical significance. p values at the bottom of the graphs show the significance of between-group comparisons and those on top of the graphs show the significance of comparisons of the changes from baseline with zero in each group separately (intercept-only analysis). An upward or downward facing arrow next to a p value indicates a statistically significant increase or decrease from baseline, respectively. Participant numbers in both groups per time point are indicated below the graphs. Comparisons of intact HIV DNA in PBMCs (**D**) and of total HIV DNA in rectal tissue (**E**) between days 0 (D0) and 84 (D84) in the intensified and control groups. Open circles depict undetectable values, assigned the values corresponding to 50% of the assay detection limits. Wilcoxon tests were used to calculate statistical significance. All p values are marked red if significant (<0.05).

The online version of this article includes the following source data and figure supplement(s) for figure 3:

**Source data 1.** Numerical data corresponding to *Figure 3A*.

**Source data 2.** Numerical data corresponding to *Figure 3B*.

**Source data 3.** Numerical data corresponding to *Figure 3C*.

**Source data 4.** Numerical data corresponding to *Figure 3D*.

**Source data 5.** Numerical data corresponding to *Figure 3E*.

**Figure supplement 1.** Longitudinal dynamics of total HIV DNA in PBMCs.

**Figure supplement 1—source data 1.** Numerical data corresponding to *Figure 3—figure supplement 1A*.

**Figure supplement 1—source data 2.** Numerical data corresponding to *Figure 3—figure supplement 1B*.

**Figure supplement 1—source data 3.** Numerical data corresponding to *Figure 3—figure supplement 1C*.

**Figure supplement 1—source data 4.** Numerical data corresponding to *Figure 3—figure supplement 1D*.

**Figure supplement 1—source data 5.** Numerical data corresponding to *Figure 3—figure supplement 1E*.

**Figure supplement 2.** Longitudinal dynamics of US HIV RNA in PBMCs.

**Figure supplement 2—source data 1.** Numerical data corresponding to *Figure 3—figure supplement 2A*.

**Figure supplement 2—source data 2.** Numerical data corresponding to *Figure 3—figure supplement 2B*.

**Figure supplement 2—source data 3.** Numerical data corresponding to *Figure 3—figure supplement 2C*.

**Figure supplement 2—source data 4.** Numerical data corresponding to *Figure 3—figure supplement 2D*.

**Figure supplement 2—source data 5.** Numerical data corresponding to *Figure 3—figure supplement 2E*.

*Figure 3 continued on next page*

*Figure 3 continued*

**Figure supplement 3.** Longitudinal dynamics of US RNA/total DNA ratio in PBMCs.

**Figure supplement 3—source data 1.** Numerical data corresponding to *Figure 3—figure supplement 3A*.

**Figure supplement 3—source data 2.** Numerical data corresponding to *Figure 3—figure supplement 3B*.

**Figure supplement 3—source data 3.** Numerical data corresponding to *Figure 3—figure supplement 3C*.

**Figure supplement 3—source data 4.** Numerical data corresponding to *Figure 3—figure supplement 3D*.

**Figure supplement 3—source data 5.** Numerical data corresponding to *Figure 3—figure supplement 3E*.

## Correlations between the measured parameters

Finally, we assessed pairwise correlations between the measured parameters at baseline, as well as between their time-weighted changes from baseline during the study (time-weighted areas under curve) and between their relative changes between days 0 and 84 of the study (*Figure 7*, *Figure 7—figure supplements 1 and 2*). At baseline, we found strong positive pairwise correlations between the levels of different plasma inflammatory cytokines, as well as between the percentages of peripheral CD8+ T cells expressing TIGIT and inflammatory cytokine levels in plasma and tissue (*Figure 7*). Of the latter, especially strong correlations were observed between CD8+TIGIT+ cell percentages and plasma or tissue levels of IL-6 (rho = 0.74 and rho = 0.71, respectively; p < 0.001). We also found positive correlations at baseline between US RNA/total DNA ratio and percentages of CD4+ T cells expressing CD38 and co-expressing CD38 and HLA-DR, markers of immune activation (rho = 0.53, p = 0.020 and rho = 0.50, p = 0.023, respectively). Furthermore, US RNA changes between days 0 and 84 positively correlated with changes in levels of expression of immune activation/exhaustion markers CD38+ on CD4+ cells and PD-1+ on CD8+ cells (rho = 0.53, p = 0.030 and rho = 0.61, p = 0.011, respectively) and of tissue inflammatory cytokine IL-1β mRNA (rho = 0.62, p = 0.0096) (*Figure 7—figure supplement 2*). These correlations were even stronger for the changes between days 0 and 84 of US RNA/total DNA ratio (CD4+CD38+: rho = 0.55, p = 0.029; CD8+PD-1+: rho = 0.73, p = 0.0018; IL-1β mRNA: rho = 0.76, p = 0.0011). This confirms the results of our earlier study, where US RNA/total DNA ratio was found to correlate with markers of immune exhaustion (*Scherpenisse et al., 2021*). Interestingly, intact, but not total, HIV DNA in PBMCs strongly negatively correlated with time of viral suppression (rho = −0.67, p = 0.0016), confirming earlier observations that intact HIV DNA decays on ART much faster than the defective (or total) HIV DNA (*Peluso et al., 2020*; *Gandhi et al., 2021*).

## Discussion

The aim of this study was to investigate whether ongoing viral replication contributes to HIV persistence under ART. To the best of our knowledge, this is the first study where ART intensification was achieved not by adding a new antiretroviral drug to the existing regimen but by increasing the dosage of a drug that was already part of the regimen pre-intensification. We evaluated the impact of doubling the dosage of DTG on HIV cellular and tissue reservoirs, immune activation, exhaustion, and inflammation in ART-suppressed PWH.

As expected, plasma and tissue DTG concentrations significantly increased between day 0 and day 84 in the intensified group but not in the control group. Accordingly, significant decreases in total HIV DNA, intact HIV DNA, US HIV RNA, and US RNA/total DNA ratio in PBMCs were observed during the study period in the intensified group but not in the control group. This strongly suggests that the pre-intensification ART regimen was not completely suppressive. The transient decrease in total DNA at day 28 may have been due to an initial decrease in newly infected cells upon ART intensification; however, at the subsequent time points, this effect was likely masked by proliferation (clonal expansion) of infected cells, mostly containing defective proviruses. In line with this, the number of intact proviruses decreased, but that of total proviruses did not change, between days 0 and 84 (*Figure 3A, D*).

In addition, intensification modestly yet significantly reduced percentages of CD4+ T cells expressing TIGIT, a T-cell exhaustion marker (*Chew et al., 2016*), and percentages of CD8+ T cells co-expressing CD38 and HLA-DR, T-cell activation markers. Interestingly, CD4+ cells expressing TIGIT have been shown to be enriched for HIV persistence in ART-treated PWH (*Fromentin et al., 2016*). Arguably, the reduction of HIV reservoir upon intensification has caused a corresponding reduction in

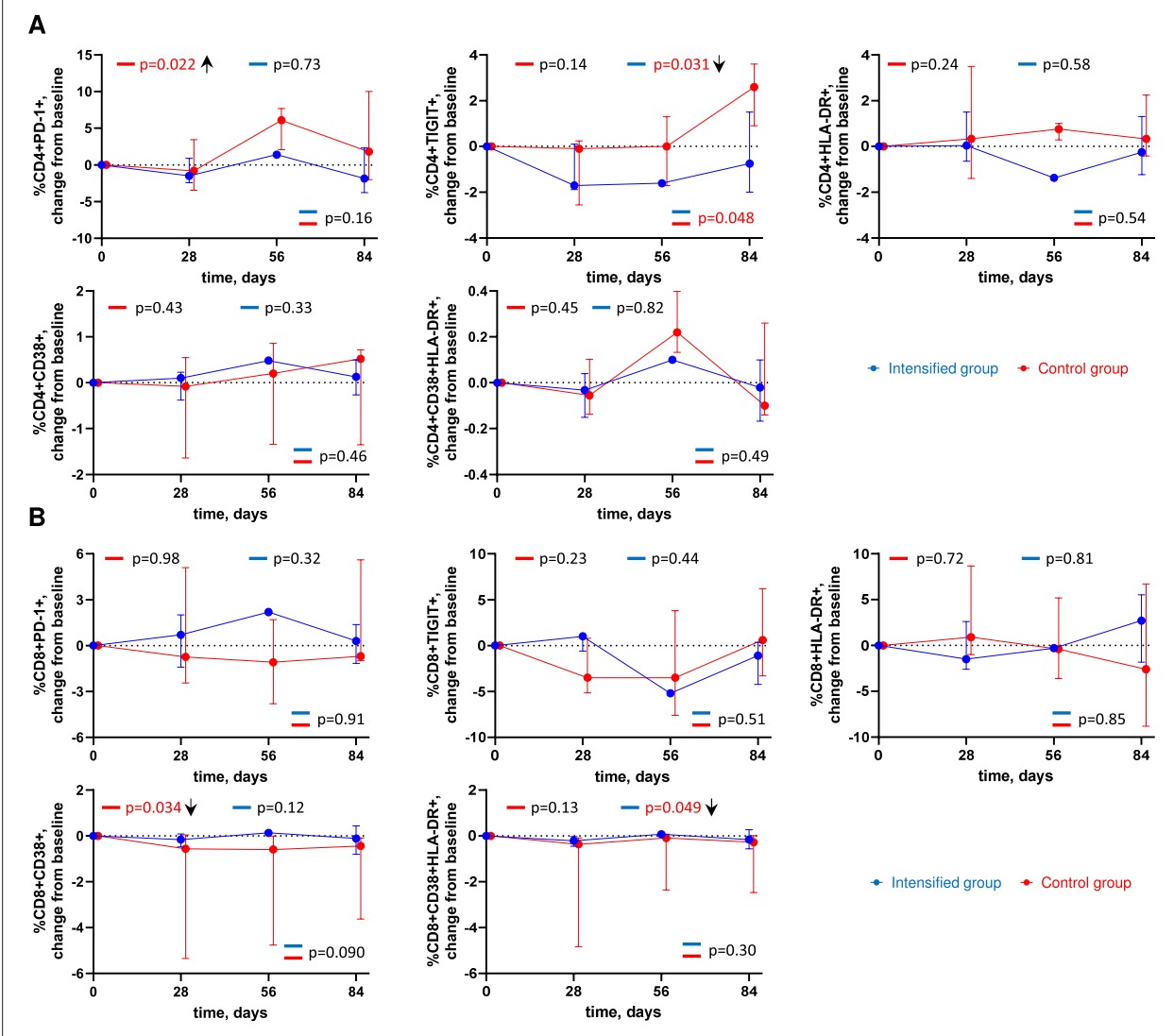

**Figure 4.** Longitudinal dynamics of cellular markers of immune activation and exhaustion. Graphs show changes of (**A**) CD4+ cell markers and (**B**) CD8+ cell markers from baseline on days 1, 28, 56, and 84 of the study in the intensified (blue) and control (red) groups. Median values and interquartile ranges (IQRs) are shown. Participant numbers: intensified group, n=10; control group, n=10. Linear mixed-effects modelling was used to calculate statistical significance. p values at the bottom of the graphs show the significance of between-group comparisons and those on top of the graphs show the significance of comparisons of the changes from baseline with zero in each group separately (intercept-only analysis). An upward or downward facing arrow next to a p value indicates a statistically significant increase or decrease from baseline, respectively. p values are marked red if significant (<0.05).

The online version of this article includes the following source data and figure supplement(s) for figure 4:

**Source data 1.** Numerical data corresponding to *Figure 4A*.

**Source data 2.** Numerical data corresponding to *Figure 4B*.

**Figure supplement 1.** Levels of cellular markers of immune activation and exhaustion.

**Figure supplement 1—source data 1.** Numerical data corresponding to *Figure 4—figure supplement 1A*.

**Figure supplement 1—source data 2.** Numerical data corresponding to *Figure 4—figure supplement 1B*.

chronic immune activation and exhaustion in this cohort. Indeed, changes in US HIV RNA levels and US RNA/total DNA ratio during the study positively correlated with changes in levels of markers of immune activation, exhaustion, and inflammation, suggesting a link between changes in HIV reservoir activity and levels of immune activation and inflammation. These results are in line with previous studies reporting decreases in immune activation markers upon ART intensification (*Yukl et al., 2010*; *Llibre et al., 2012*; *Vallejo et al., 2012*; *Gutiérrez et al., 2011*).

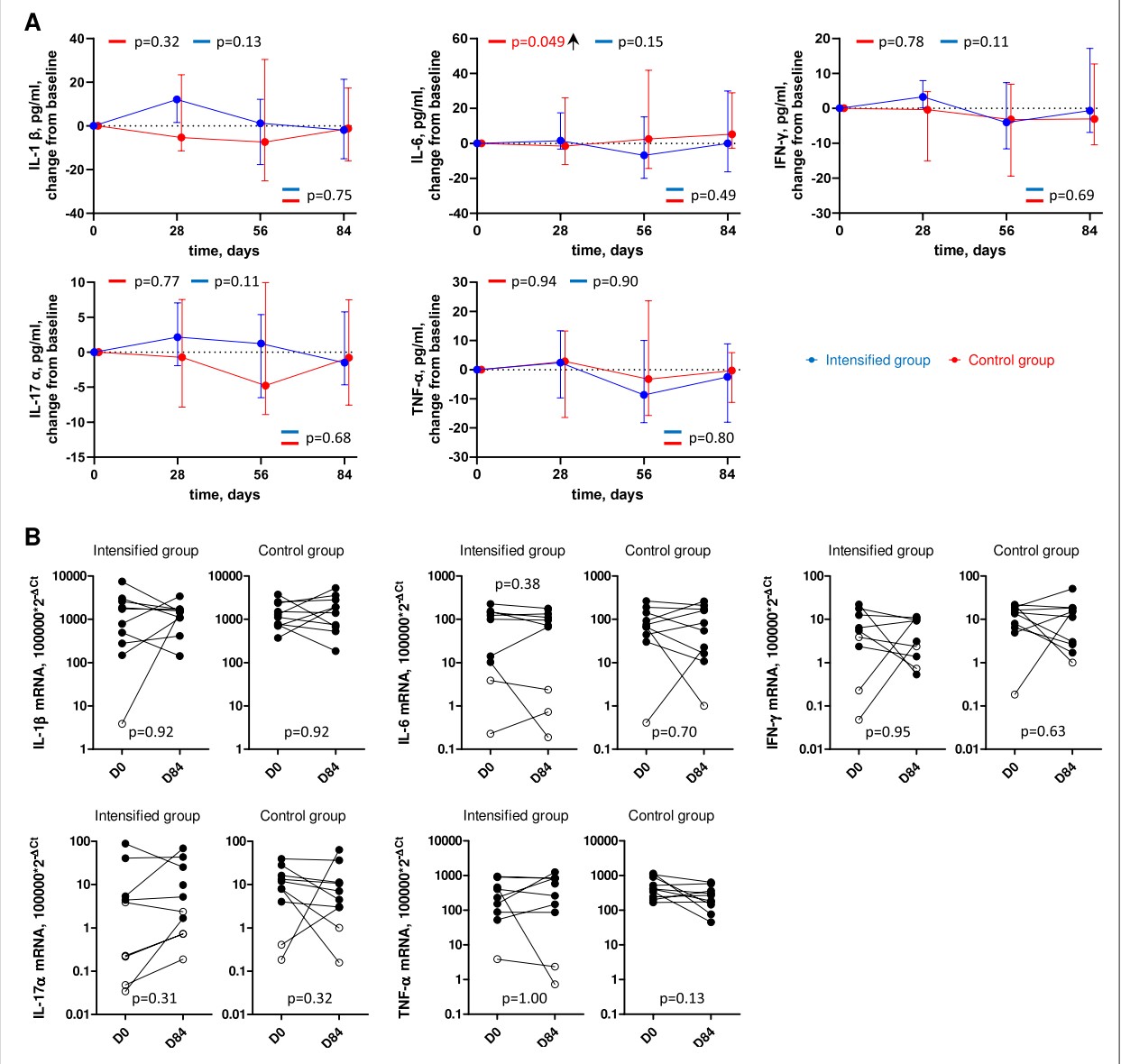

**Figure 5.** Longitudinal dynamics of inflammation markers in plasma and tissue. (**A**) Plasma inflammation markers. Graphs show changes of markers from baseline on days 1, 28, 56, and 84 of the study in the intensified (blue) and control (red) groups. Median values and interquartile ranges (IQRs) are shown. Participant numbers: intensified group, n=10; control group, n=10. Linear mixed-effects modelling was used to calculate statistical significance. p values at the bottom of the graphs show the significance of between-group comparisons and those on top of the graphs show the significance of comparisons of the changes from baseline with zero in each group separately (intercept-only analysis). An upward or downward facing arrow next to a p value indicates a statistically significant increase or decrease from baseline, respectively. p values are marked red if significant (<0.05). (**B**) Tissue inflammation markers. Markers were compared between days 0 (D0) and 84 (D84) in the intensified and control groups. Open circles depict undetectable values, assigned the values corresponding to 50% of the assay detection limits. Wilcoxon tests were used to calculate statistical significance.

The online version of this article includes the following source data and figure supplement(s) for figure 5:

**Source data 1.** Numerical data corresponding to *Figure 5A*.

**Source data 2.** Numerical data corresponding to *Figure 5B*.

**Figure supplement 1.** Levels of inflammation markers in plasma in intensified (blue) and control (red) groups.

**Figure supplement 1—source data 1.** Numerical data corresponding to *Figure 5—figure supplement 1*.

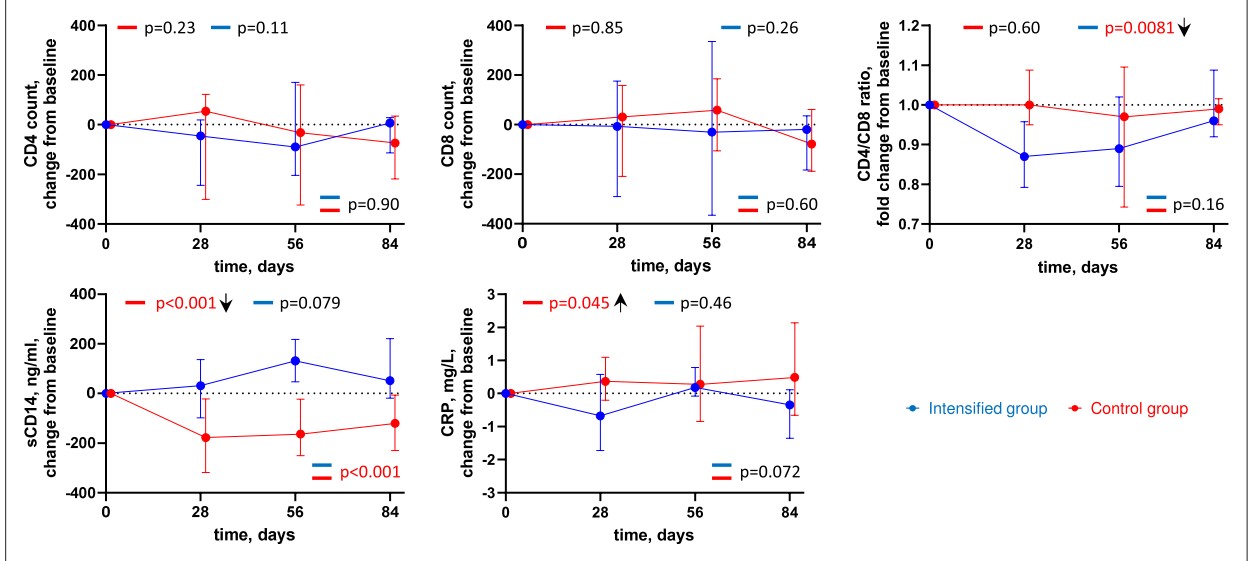

**Figure 6.** Longitudinal dynamics of clinical markers. Graphs show changes of markers from baseline on days 1, 28, 56, and 84 of the study in the intensified (blue) and control (red) groups. Median values and interquartile ranges (IQRs) are shown. Participant numbers: intensified group, n=10; control group, n=10. Linear mixed-effects modelling was used to calculate statistical significance. p values at the bottom of the graphs show the significance of between-group comparisons and those on top of the graphs show the significance of comparisons of the changes from baseline with zero in each group separately (intercept-only analysis). An upward or downward facing arrow next to a p value indicates a statistically significant increase or decrease from baseline, respectively. p values are marked red if significant (<0.05).

The online version of this article includes the following source data and figure supplement(s) for figure 6:

**Source data 1.** Numerical data corresponding to *Figure 6*.

**Figure supplement 1.** Levels of clinical markers in intensified (blue) and control (red) groups.

**Figure supplement 1—source data 1.** Numerical data corresponding to *Figure 6—figure supplement 1*.

In contrast, in the control group, we measured significant longitudinal increases in total HIV DNA, percentages of CD4+ T cells expressing PD-1, another T-cell exhaustion marker, and in the plasma levels of IL-6, an inflammatory cytokine, and of CRP, another marker of inflammation. Residual HIV replication, among other factors, could have contributed to these effects in the control group. However, in the control group, we also observed a reduction in the percentages of CD8+ T cells co-expressing a T-cell activation marker CD38 and a strong reduction in the plasma levels of sCD14. One possible explanation for these effects can be increased adherence of the participants to their ART regimens during the study.

Two previous studies have measured a transient increase in HIV episomal DNA (2-LTR circles) upon ART intensification with raltegravir, an INSTI (*Buzón et al., 2010*; *Hatano et al., 2013*). As 2-LTR circles are a by-product of HIV integration and are expected to accumulate if integration is blocked, this was interpreted as strong evidence of new infections pre-intensification. However, another study did not observe any change in 2-LTR circles upon ART intensification with DTG (*Rasmussen et al., 2018*). In our study, despite repeated attempts using different assays, we could only detect 2-LTR circles in a small minority of participants (<5%) (data not shown). Notably, the effects of intensification on 2-LTR circles in these previous studies were much more pronounced in participants who had been treated with a PI-based ART regimen pre-intensification (*Buzón et al., 2010*; *Hatano et al., 2013*). As PI-based regimens have been shown to be less suppressive than other ART regimens (*Pasternak et al., 2021*; *Darcis et al., 2020b*; *Darcis et al., 2020a*; *Hatano et al., 2011*), they might have allowed for a higher level of new infections and consequently of 2-LTR circles in these studies.

In contrast, in our study, participants had been treated with a DTG-based regimen for at least 2 years and had had a suppressed plasma viral load (<50 copies/ml) for at least 2 years 11 months pre-intensification. Plausibly, the lack of detectable 2-LTR circles in our participants reflects the high potency of their ART regimens and a long period of virological suppression before the study. Remarkably, the intensification revealed that even this highly potent regimen was not completely suppressive.

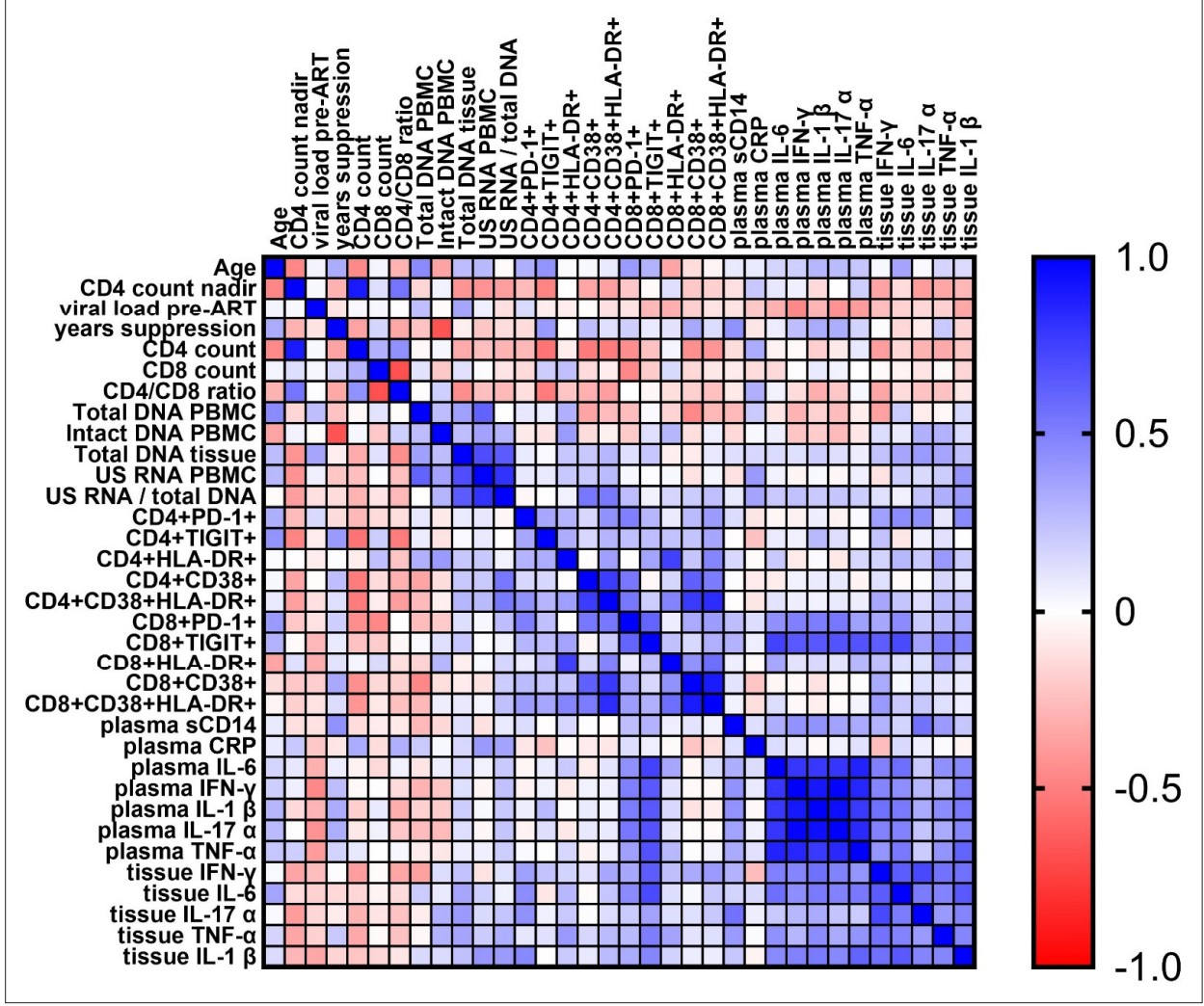

**Figure 7.** Spearman correlogram of baseline parameters. A heat map is used to indicate the strengths of associations between parameters. Red indicates a negative correlation, and blue indicates a positive correlation.

The online version of this article includes the following source data and figure supplement(s) for figure 7:

**Source data 1.** Numerical data corresponding to *Figure 7*.

**Figure supplement 1.** Spearman correlogram of time-weighted changes from baseline.

**Figure supplement 1—source data 1.** Numerical data corresponding to *Figure 7—figure supplement 1*.

**Figure supplement 2.** Spearman correlogram of changes between days 0 and 84.

**Figure supplement 2—source data 1.** Numerical data corresponding to *Figure 7—figure supplement 2*.

Thus, our results emphasize that 2-LTR circles are not a very sensitive marker of new infections due to their relative scarcity in infected cells. 2-LTR circles represent only ~10% of total episomal DNA (*Butler et al., 2001*) and are not expected to be present in every newly infected cell. In contrast, a newly infected CD4+ T lymphocyte can contain hundreds to thousands of US HIV RNA copies on the peak of infection (*Hockett et al., 1999*). Therefore, a change in the US RNA level upon ART intensification is a much more sensitive indicator of new infections than that of 2-LTR circles, as confirmed by this study.

Due to low penetration of antiretroviral drugs into tissue sites and anatomic sanctuaries (*Fletcher et al., 2014*; *Fletcher et al., 2022*; *Estes et al., 2017*; *Rothenberger et al., 2019*; *Lee et al., 2020*), ART pressure on HIV replication is generally expected to be more relaxed in tissues than in peripheral blood. However, we did not observe any effect of the intensification on total HIV DNA in the rectal tissue. Although DTG concentration in the rectal tissue did increase upon intensification, it remained much lower than in the peripheral blood. This possibly contributed to the lack of detectable effect of

intensification on the HIV DNA in tissue. Alternatively, intensification with DTG, an INSTI, had no effect in the tissue due to the preponderance of abortive infection events in lymphoid tissues that occur by cell-to-cell contact and are terminated prior to integration (*Galloway et al., 2015*; *Doitsh et al., 2010*). It is also possible that, similarly to the peripheral blood, the effect of the intensification on the reservoir in tissue was masked by the clonal expansion of defective proviruses by day 84 of the study.

The majority of previous studies that used INSTIs (raltegravir or DTG) as intensification drugs added it as a fourth drug using the standard dosage. For example, the DTG dosage in the study of Rasmussen et al. was 50 mg daily (*Rasmussen et al., 2018*). In our study, we used a twice-higher dosage (100 mg daily). This DTG dosage is commonly used in clinical practice. Indeed, in PWH with confirmed or clinically suspected resistance to the integrase inhibitor class, the recommended dosage of DTG is 50 mg twice daily (https://www.cbip.be/fr/chapters/12?frag=10868). Based on our results (and pending the confirmation by larger studies), an increased dosage of DTG might be warranted in the future as a part of a standard regimen even in PWH without resistance. However, this has to be approached with caution as intensification has caused a transient decrease in the CD4/CD8 ratio in our study.

The main strength of our study is the measurement of a large number of virological and immunological parameters that allowed us to observe a significant impact of intensification on both virus and host. Our study is the first to observe significant reductions in as many as four HIV markers following ART intensification. Moreover, we measured plasma and tissue concentrations of DTG, allowing us to directly demonstrate their increase upon intensification. Our results are in line with some earlier studies that reported a decrease in cell-associated HIV RNA or a transient increase in 2-LTR circles upon intensification (*Buzón et al., 2010*; *Yukl et al., 2010*; *Hatano et al., 2013*). However, they are in contrast with other studies, in which treatment intensification did not affect HIV markers (*Dinoso et al., 2009*; *McMahon et al., 2010*; *Gandhi et al., 2010*). As this is the first ART intensification study to increase the dosage of an existing antiretroviral drug instead of adding a new drug, the results of this study and those of previous intensification studies cannot be directly compared.

Some limitations can be highlighted. First, we included twenty PWH (10 participants per arm). Although this small sample size does somewhat limit the robustness of the conclusions, we would like to emphasize that in this cohort, we performed a very detailed analysis, longitudinally measuring 30 biomarkers at five time points during ART. Secondly, despite the randomization, the groups were still not perfectly balanced as levels of total HIV DNA in PBMCs were by chance higher at baseline in the intensified group. However, this is unlikely to have affected the conclusions of the study because (1) this study has a within-subject (repeated-measures) design, with the longitudinal change of a parameter within the same participant during the study being the main outcome, and (2) this difference most probably reflects a higher number of infected cells, and not a higher level of residual replication, at baseline in the intensified group. No significant differences between the groups were found at baseline in the levels of the other four virological markers we measured (US RNA in PBMCs, US RNA/total DNA ratio, intact DNA in PBMCs, and total DNA in tissue), as well as between the antiretroviral drug concentrations in plasma or in tissue. The lack of blinding and placebo control, the predominantly male study population, and the absence of post-intervention follow-up represent additional limitations of our study. Moreover, these findings should be considered exploratory and hypothesis-generating, warranting confirmation in larger, placebo-controlled, and blinded trials. Nevertheless, we believe that the convergence of the effect of intensification on multiple reservoir markers in the same direction indicates a potentially meaningful biological signal that merits further investigation.

In conclusion, doubling the DTG dosage in PWH who had been suppressed on DTG-containing ART for a number of years resulted in a reduction in the levels of four HIV reservoir markers in peripheral blood, as well as in markers of T-cell activation and exhaustion. If confirmed in larger clinical trials, these results could have an impact on the clinical management of PWH. Moreover, if ongoing viral replication does indeed replenish HIV reservoirs over time, then improving ART regimens should be a necessary part of the curative strategies.

# Methods

**Key resources table**

| Reagent type (species) or resource | Designation | Source or reference | Identifiers | Additional information |
|---|---|---|---|---|
| Antibody | APC/Fire 810 anti-human CD3 (Mouse monoclonal) | Sony Biotechnology | Cat# 2324290 | 1:25 |
| Antibody | PE/Fire 700 anti-human CD4 (Mouse monoclonal) | Sony Biotechnology | Cat# 2323330 | 1:25 |
| Antibody | PerCP anti-human CD8a (Mouse monoclonal) | Sony Biotechnology | Cat# 2104610 | 1:25 |
| Antibody | PE Mouse anti-human CD279 (PD-1) (Mouse monoclonal) | BD Biosciences | Cat# 560795; RRID:AB_2033989 | 1:25 |
| Antibody | BV421 Mouse anti-human TIGIT (Mouse monoclonal) | BD Biosciences | Cat# 747844; RRID:AB_2872307 | 1:25 |
| Antibody | BB515 Mouse anti-human HLA-DR (Mouse monoclonal) | BD Biosciences | Cat# 564516; RRID:AB_2732846 | 1:25 |
| Antibody | BV711 Mouse anti-human CD38 (Mouse monoclonal) | BD Biosciences | Cat# 563965; RRID:AB_2738516 | 1:25 |
| Commercial assay or kit | DNA-free DNA Removal Kit | Thermo Fisher Scientific | Cat# AM1906 | |
| Commercial assay or kit | TaqMan β-Actin Detection Reagents | Thermo Fisher Scientific | Cat# 401846 | |
| Commercial assay or kit | TaqMan Ribosomal RNA Control Reagents | Thermo Fisher Scientific | Cat# 4308329 | |
| Commercial assay or kit | Puregene Cell Kit | QIAGEN | Cat# 158043 | |
| Commercial assay or kit | ReliaPrep gDNA Tissue Miniprep System | Promega | Cat# A2051 | |
| Commercial assay or kit | ProcartaPlex Human Inflammation Panel, 20plex | Thermo Fisher Scientific | Cat# EPX200-12185-901 | |
| Commercial assay or kit | Human CD14 ELISA Kit – Quantikine | R&D Systems | Cat# DC140 | |
| Chemical compound, drug | Platinum Quantitative PCR SuperMix-UDG | Thermo Fisher Scientific | Cat# 11730-025 | |
| Chemical compound, drug | SuperScript III reverse transcriptase | Thermo Fisher Scientific | Cat# 18080-085 | |
| Chemical compound, drug | Random primers | Thermo Fisher Scientific | Cat# 48190-011 | |
| Chemical compound, drug | RNaseOUT Recombinant Ribonuclease Inhibitor | Thermo Fisher Scientific | Cat# 10777-019 | |
| Chemical compound, drug | ddPCR Supermix for Probes (No dUTP) | Bio-Rad | Cat# 1863024 | |
| Chemical compound, drug | *Bgl*I restriction enzyme | Thermo Fisher Scientific | Cat# ER0071 | |
| Chemical compound, drug | TaqMan Gene Expression Assay, IL-1β | Thermo Fisher Scientific | Cat# 4331182 | Hs01555410_m1 |
| Chemical compound, drug | TaqMan Gene Expression Assay, IL-6 | Thermo Fisher Scientific | Cat# 4331182 | Hs00174131_m1 |
| Chemical compound, drug | TaqMan Gene Expression Assay, IFN-γ | Thermo Fisher Scientific | Cat# 4331182 | Hs00989291_m1 |
| Chemical compound, drug | TaqMan Gene Expression Assay, IL-17α | Thermo Fisher Scientific | Cat# 4331182 | Hs00174383_m1 |

*Continued on next page*

*Continued*

| Reagent type (species) or resource | Designation | Source or reference | Identifiers | Additional information |
|---|---|---|---|---|
| Chemical compound, drug | TaqMan Gene Expression Assay, TNF-α | Thermo Fisher Scientific | Cat# 4331182 | Hs00174128_m1 |
| Chemical compound, drug | TaqMan Gene Expression Assay, GAPDH | Thermo Fisher Scientific | Cat# 4331182 | Hs02758991_g1 |
| Software, algorithm | Prism 10.2.0 | GraphPad Software | https://www.graphpad.com/ RRID:SCR_002798 | Statistics |
| Software, algorithm | IBM SPSS Statistics 28.0.1.0 | IBM | https://www.ibm.com/products/spss-statistics RRID:SCR_016479 | Statistics |
| Software, algorithm | QuantaSoft 1.7.4 | Bio-Rad | RRID:SCR_025696 | ddPCR data analysis |
| Software, algorithm | Rotor-Gene 2.3.5 | QIAGEN | RRID:SCR_015740 | qPCR data analysis |
| Software, algorithm | FlowJo 10.8.1 | Becton Dickinson | https://www.flowjo.com/ RRID:SCR_008520 | Flow cytometry data analysis |

## Study design and participants

This was a phase 2 open-label, interventional, monocentric, randomized, and controlled clinical trial performed at the Hospital University of Liège (Belgium) in which 20 HIV-infected adults were enrolled. Eligible participants had been treated with a combination of 50 mg DTG, 600 mg ABC, and 300 mg 3TC for more than 2 years, and the purpose of this study was to assess the impact of DTG intensification (an additional daily dose of 50 mg) compared to a control group that continued with the above regimen. This study was a proof-of-concept trial designed to reveal biological effects of ART intensification, and the primary outcome was defined as 'to evaluate the impact of treatment intensification at the level of total and replication-competent reservoir in blood and in tissues', with a time frame of 3 months. Participants were required to have a plasma viral load suppressed below 20 HIV RNA copies/ml during the 12 months before screening (allowing 'blips') and an absolute CD4+ T lymphocyte count above 200 cells/mm$^3$. One participant out of 20 had an isolated 'blip' of 115 copies/ml 9 months before the study initiation. The detailed inclusion and exclusion criteria can be found in the appendix (Appendix 1). The study was approved by the Liège University Hospital-Faculty Ethics Committee (Comité d'Éthique Hospitalo-Facultaire Universitaire de Liège, 2018/228). Prior to inclusion in the study, each participant had dated and signed an informed consent form. https://clinicaltrials.gov/ identifier: NCT05351684.

Historical plasma HIV RNA measurements, CD4+ T cell counts, and treatment data were retrieved from the outpatient medical records. The duration of continuous virological suppression was calculated as the duration of the latest period with undetectable plasma HIV RNA prior to the study initiation, allowing isolated 'blips' of 50–999 copies/ml. The duration of cumulative suppression was calculated by adding together all such periods of continuous suppression.

## Clinical procedures

Each participant was randomly assigned to the intensified group or to the control group. This study lasted 84 days in which five visits were programmed (days 0, 1, 28, 56, and 84). One visit prior to inclusion was added to perform a blood test to ensure that the participant met the inclusion/exclusion criteria for the study. The blood test included blood and platelet count, electrolyte panel, creatinine, LDH counts, liver tests, coagulation tests, CD4 and CD8 lymphocyte counts and a plasma viral load. β-HCG levels were measured in women at each visit to ensure that they were not pregnant. All participants completed the 12-week-long study and no individuals were lost to follow-up.

Safety and tolerability evaluations were conducted on day 28 and day 84 and included a clinical examination and routine laboratory testing (liver function tests, kidney function, and complete blood count). Medication adherence was also monitored through pill counts performed by the study nurses.

No virological blips above 50 copies/mL were observed and no adverse events were reported by participants during the 3-month intensification period. Although CPK levels were not included in the

routine biological monitoring, no participant reported muscle pain or other symptoms suggestive of muscle toxicity.

Blood samples were collected in 4 EDTA tubes (~40 ml) at days 0, 1, 28, 56, and 84 and directly sent to the AIDS reference laboratory of Liège University Hospital for PBMC isolation. PBMC samples were stored at –150°C for measurements of the HIV reservoir and immunological parameters. Plasma samples were used to measure soluble inflammation markers. Ten flash-frozen rectal biopsies were collected on days 0 and 84 during a routine gastroenterology visit. The cryotubes containing the biopsies were directly placed in liquid nitrogen and stored at –150°C.

## PBMC isolation

PBMCs were isolated from whole blood by density gradient centrifugation using Ficoll-Paque Plus. The blood samples were collected in EDTA tubes and centrifuged at $1200 \times g$ for 10 min (acceleration 9, brake 9). The plasma was removed and stored at –150°C. The blood was then well homogenized with RPMI 1640, gently transferred in a Falcon tube that contained 12,5 ml of Ficoll, and centrifuged at $700 \times g$ for 20 min (acceleration 6, brake 0). The PBMC layer was removed, diluted in 50 ml RPMI 1640, and centrifuged at $300 \times g$ for 7 min (acceleration 9, brake 9). The supernatant was removed, and the process was repeated two more times, completing three washes. The pellet was resuspended, and the cells were counted. Foetal bovine serum (FBS) was added to the re-suspended pellet to obtain $10 \times 10^6$ cells/500 µl. For cryopreservation, 500 µl of freeze mixture (10% DMSO in FBS) was added to 500 µl of the cell suspension in the cryotubes. Cryotubes were slowly frozen in the Mr. Frosty freezing container at –80°C and transferred to –150°C the next day.

## Quantification of total HIV DNA and cell-associated US HIV RNA in PBMCs

Total nucleic acids were extracted from $2 \times 10^6$ PBMCs using Boom isolation method (*Boom et al., 1990*). Total HIV DNA was measured by qPCR using the previously described primer/probe combination that amplifies the HIV packaging signal ($\Psi$) region (*Bruner et al., 2019*). Extracted cellular RNA was treated with DNase (DNA-free kit; Thermo Fisher Scientific) to remove genomic DNA that could interfere with the quantitation and reverse transcribed into cDNA using random primers and SuperScript III reverse transcriptase (all from Thermo Fisher Scientific). To quantify cell-associated US HIV RNA, this cDNA was pre-amplified using primer pair $\Psi$_F (*Bruner et al., 2019*) and HIV-FOR (*Malnati et al., 2008*). The product of this PCR was used as template for a seminested qPCR with the $\Psi$ primer/probe combination (*Bruner et al., 2019*). HIV DNA or RNA copy numbers were determined using a 7-point standard curve with a linear range of more than 5 orders of magnitude that was included in every qPCR run and normalized to the total cellular DNA (by measurement of β-actin DNA) or RNA (by measurement of 18S ribosomal RNA) inputs, respectively, as described previously (*Pasternak et al., 2009*). Non-template control wells were included in every qPCR run and were consistently negative. Total HIV DNA and US RNA were detectable in 90.7% and 66.7% of the samples, respectively. Undetectable measurements of US RNA or total DNA were assigned the values corresponding to 50% of the corresponding assay detection limits, with a maximum of 25 copies/µg total RNA or 50 copies/$10^6$ PBMCs, respectively. The detection limits depended on the amounts of the normalizer (input cellular DNA or RNA) and therefore differed among samples. Measurements with low input cellular DNA or RNA and undetectable total HIV DNA or US RNA ($n = 2$ and $n = 3$, respectively) were excluded from the analysis. HIV transcription levels per provirus (US RNA/total DNA ratios) were calculated taking into account that $10^6$ PBMCs contain 1 µg of total RNA (*Fischer et al., 1999*).

## Quantification of intact HIV DNA in PBMCs

Intact HIV DNA was quantified on days 0 and 84 of the study by the IPDA (*Bruner et al., 2019*). In brief, genomic DNA was isolated from $4 \times 10^6$ PBMCs using Puregene Cell Kit (QIAGEN Benelux B.V.) according to the manufacturer's instructions and digested with *Bgl*I restriction enzyme (Thermo Fisher Scientific) as described previously (*Levy et al., 2021*). Notably, only a small minority (<8%) of HIV clade B sequences contain *Bgl*I recognition sites between $\Psi$ and *env* amplicons; therefore, *Bgl*I digestion is not expected to substantially influence the IPDA output, while improving the assay sensitivity by increasing the genomic DNA input into a ddPCR reaction (*Levy et al., 2021*). After desalting by ethanol precipitation, genomic DNA was subjected to two separate multiplex droplet digital PCR

(ddPCR) assays: one targeting HIV $\Psi$ and *env* regions using primers and probes described previously, including the unlabelled *env* competitor probe to exclude hypermutated sequences (*Bruner et al., 2019*), and one targeting the cellular *RPP30* gene, which was measured to correct for DNA shearing and to normalize the intact HIV DNA to the cellular input. The *RPP30* assay amplified two regions, with amplicons located at exactly the same distance from each other as HIV $\Psi$ and *env* amplicons. The first region was amplified using a forward primer 5'-AGATTTGGACCTGCGAGCG-3', a reverse primer 5'-GAGCGGCTGTCTCCACAAGT-3', and a fluorescent probe 5'-FAM-TTCTGACCTGAAGGCT CTGCGCG-BHQ1-3' (*Luo et al., 2005*). The second region was amplified using a forward primer 5'-AGAGAGCAACTTCTTCAAGGG-3', a reverse primer 5'-TCATCTACAAAGTCAGAACATCAGA-3', and a fluorescent probe 5'-HEX-CCCGGCTCTATGATGTTGTTGCAGT-BHQ1-3'. The ddPCR conditions were as described previously (*Bruner et al., 2019*) with some minor amendments: we used 46 cycles of denaturation/annealing/extension and the annealing/extension temperature was 60°C. Intact HIV DNA was detectable in 79.5% of the samples. Undetectable measurements of intact HIV DNA were assigned the values corresponding to 50% of the assay detection limits. The detection limits depended on the amounts of the normalizer (input cellular DNA) and therefore differed among samples. For two participants, in whom intact HIV DNA on day 0 was undetectable due to a technical issue, we included day 1 values in the analysis. Excluding these two participants did not change the conclusions (data not shown).

## Quantification of total HIV DNA in rectal biopsies

Prior to total HIV DNA isolation, rectal biopsies were homogenized with the TissueLyser (QIAGEN Benelux B.V.). Total HIV DNA was isolated using ReliaPrep gDNA Tissue Miniprep System (Promega) according to the manufacturer's instructions. Total HIV DNA in rectal biopsies was quantified using exactly the same method as total HIV DNA in PBMCs (see above). Total HIV DNA was detectable in 92.5% of the rectal tissue samples. Undetectable measurements of total HIV DNA were assigned the values corresponding to 50% of the corresponding assay detection limits. The detection limits depended on the amounts of the normalizer (input cellular DNA) and therefore differed among samples.

## Quantification of immune activation and exhaustion

Flow cytometry was used to quantify surface markers of immune activation and exhaustion (CD38, HLA-DR, PD-1, and TIGIT) on CD4+ and CD8+ T cells. Cryopreserved PBMCs ($10 \times 10^6$ cells) were thawed and stained with a fixable viability dye Zombie NIR (Sony), and the following antibodies: CD3-APC/Fire810, CD4-PE/Fire700, CD8a-PercP (all from Sony), PD-1-PE, TIGIT-BV421, HLA-DR-BB515, and CD38-BV711 (all from BD Biosciences). Cells were then washed, fixed in 1% paraformaldehyde, and run on a FACS SONY ID7000. Analysis was performed using FlowJo software (v10.8.1). Cells were gated according to morphological parameters (forward and side scatter) and viability. The complete gating strategy can be found in the appendix (Appendix 3). To exclude the effect of inter-assay variation on the data, samples from each participant were always analysed together.

## Quantification of systemic and tissue inflammation

To quantify systemic inflammation, the levels of five plasma inflammatory cytokines (IFN-γ, IL-1β, IL-6, IL-17α, and TNF-α) were measured on MAGPIX (Luminex) using the ProcartaPlex Human Inflammation Panel, 20plex (Thermo Fisher Scientific). sCD14 was quantified by Human CD14 Quantikine ELISA Kit (R&D Systems). CRP was measured by high-sensitivity C-reactive protein (hs-CRP) test.

To measure tissue inflammation, we determined rectal tissue levels of mRNAs encoding five inflammatory cytokines (IL-1β, IL-6, IFN-γ, IL-17 α, and TNF-α). Rectal tissue biopsies were manually homogenized and lysed in buffer L6 (*Boom et al., 1990*) and RNA was extracted using Boom isolation method (*Boom et al., 1990*). Extracted RNA was treated with DNase (DNA-free kit; Thermo Fisher Scientific) and reverse transcribed using random primers and SuperScript III reverse transcriptase (all from Thermo Fisher Scientific). Cytokine mRNAs were quantified using TaqMan Gene Expression Assays (all from Thermo Fisher Scientific): IL-1β – Hs01555410_m1, IL-6 – Hs00174131_m1, IFN-γ – Hs00989291_m1, IL-17α – Hs00174383_m1, and TNF-α – Hs00174128_m1. Levels of cytokine mRNAs were normalized to that of *GAPDH* mRNA, measured using a TaqMan Gene Expression Assay Hs02758991_g1.

## Quantification of antiretroviral drug concentrations

Briefly, DTG was extracted from plasma samples via methanolic protein precipitation and from tissue samples via tissue homogenization, followed by protein precipitation. A DTG standard calibration curve over the range 20.0–5000 ng/ml and stable isotope labelled internal standard were used to generate an analyte/internal standard response. Analyte/internal standard response was detected by liquid chromatography–tandem mass spectrometry (LCMS) based on previously described methods (*Fletcher et al., 2014*; *Rothenberger et al., 2019*; *Fletcher et al., 2020*). Tissue penetration ratios of DTG were calculated as the ratios of DTG concentrations in tissue (as ng/ml after conversion from ng/g by dividing the values in ng/g by 1.06; *Dyavar et al., 2021*) to the concentrations in plasma (ng/ml). Lamivudine (3TC) concentrations were determined with an LC–MS method similar to our previously described procedures (*Delahunty et al., 2006*). Differences are as follows: 3TC stock and standard working solutions were prepared in methanol and demonstrated similar stability. The MS used was a triple-quadrupole AB Sciex 5500 series monitoring positive ion transitions 230.1 -> 112.1 (3TC) and 233.1 -> 115.1 (stable labelled internal standard). The mobile phase was 12.5% acetonitrile and 0.1% formic acid in deionized water. No source rinsing mobile phase was needed. The limit of quantitation was 30 ng/ml and the upper bound was 6000 ng/ml. Total accuracy ranged from 94.3% to 96.9% for. Within-day coefficients of variation were below 5%, between-day coefficients of variation were below 5%. Stability of 3TC in human plasma was also similar.

## Statistical analysis

Baseline parameters were compared using Mann–Whitney tests for continuous variables and Fisher's exact tests for categorical variables. Virological parameters were $log_{10}$-transformed before analysis. Longitudinal dynamics of the measured parameters was modelled using linear mixed-effects analysis. This analysis was used both to compare the longitudinal changes from baseline between the intensified and the control groups and to compare the changes from baseline with zero in each of these groups separately (intercept-only analysis). As a sensitivity analysis, we performed a comparison between models that do and do not include time as a covariate. Including time in the models did not change the conclusions (*Appendix 2—tables 1 and 2*). Paired Wilcoxon tests were used for within-group comparisons of parameters between days 0 and 84. For these comparisons, pairs where both values were undetectable were excluded from the analysis. Time-weighted changes from baseline were calculated by dividing the areas under curve (computed using the trapezoid rule) by the time period of the study. Time-weighted changes from baseline were not calculated if the baseline value and one or more of the subsequent values were undetectable. Time-weighted changes from baseline and ratios of the parameters between different study days were compared between the groups using Mann–Whitney tests. Spearman tests were used to produce correlation matrices of the parameters at baseline, as well as of time-weighted changes from baseline and of changes between day 0 and day 84 of the measured parameters. Changes of the parameters from baseline, as well as US RNA/total DNA ratios, were not calculated if both values in a pair were undetectable. Data were analysed using Prism 10.2.0 (GraphPad Software) or IBM SPSS Statistics 28.0.1.0. All statistical tests were two-sided, and p values of <0.05 were considered statistically significant.

## Acknowledgements

We are grateful to Marine Hubert and Raquel Mateus for technical assistance in PBMC isolation. We acknowledge the participation and commitment of the trial participants who made the study possible. CFL is a FRIA grantee of the Belgian Fund for Scientific Research (Fonds de la Recherche Scientifique – FNRS) and is supported by the Fondation Léon Fredericq. This work was supported in part by grant R01-AI124965 (to CVF) from the US National Institutes of Health. AOP acknowledges grant support from amfAR, The Foundation for AIDS Research (grant no. 1110680-77-RPRL), and from Partnership NWO-Dutch AIDS Fonds 'HIV cure for everyone' (grant no. KICH2.V4P.AF23.001). GD is supported by the Belgian Fund for Scientific Research (Fonds de la Recherche Scientifique – FNRS).

## Additional information

### Funding

| Funder | Grant reference number | Author |
| --- | --- | --- |
| Fonds De La Recherche Scientifique - FNRS | | Céline Fombellida-Lopez Gilles Darcis |
| Fonds Léon Fredericq | | Céline Fombellida-Lopez |
| National Institutes of Health | R01-AI124965 | Courtney V Fletcher |
| amfAR, The Foundation for AIDS Research | 1110680-77-RPRL | Alexander O Pasternak |
| Nederlandse Organisatie voor Wetenschappelijk Onderzoek | KICH2.V4P.AF23.001 | Alexander O Pasternak |

The funders had no role in study design, data collection, and interpretation, or the decision to submit the work for publication.

### Author contributions

Céline Fombellida-Lopez, Resources, Data curation, Formal analysis, Validation, Investigation, Writing – original draft, Project administration, Writing – review and editing; Aurelija Valaitienė, Patricia Dellot, Aurélie Ladang, Fabrice Susin, Catherine Reenaers, Investigation; Lee Winchester, Investigation, Methodology, Writing – review and editing; Nathalie Maes, Formal analysis, Methodology; Céline Vanwinge, Supervision, Investigation, Writing – review and editing; Etienne Cavalier, Dolores Vaira, Marie-Pierre Hayette, Supervision; Michel Moutschen, Supervision, Writing – review and editing; Courtney V Fletcher, Supervision, Investigation, Methodology, Writing – review and editing; Alexander O Pasternak, Resources, Data curation, Formal analysis, Supervision, Funding acquisition, Investigation, Methodology, Writing – original draft, Writing – review and editing; Gilles Darcis, Conceptualization, Resources, Supervision, Funding acquisition, Investigation, Writing – original draft, Project administration, Writing – review and editing

### Author ORCIDs

Céline Fombellida-Lopez ⓘ https://orcid.org/0000-0001-9401-7150
Courtney V Fletcher ⓘ https://orcid.org/0000-0002-3703-7849
Alexander O Pasternak ⓘ https://orcid.org/0000-0002-4097-4251
Gilles Darcis ⓘ https://orcid.org/0000-0001-8192-1351

### Ethics

Clinical trial registration ClinicalTrials.gov identifier: NCT05351684.
The study was approved by the Liège University Hospital-Faculty Ethics Committee (Comité d'Éthique Hospitalo-Facultaire Universitaire de Liège, 2018/228). Prior to inclusion in the study, each participant had dated and signed an informed consent form.

Reviewer #1 (Public review): https://doi.org/10.7554/eLife.106931.3.sa1
Reviewer #2 (Public review): https://doi.org/10.7554/eLife.106931.3.sa2
Reviewer #3 (Public review): https://doi.org/10.7554/eLife.106931.3.sa3
Author response https://doi.org/10.7554/eLife.106931.3.sa4

## Additional files

### Supplementary files
MDAR checklist

### Data availability
Source data for all figures and tables is uploaded.

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

## Appendix 1

### Inclusion and exclusion criteria

Inclusion criteria

- Men or women of at least 18 years of age with a confirmed HIV infection.
- The patient must have been on ART for more than 24 months with stable treatment for at least 6 months and a viral load consistently below 20 copies/ml during the 12 months prior to screening.
- The patient should have a CD4+ T lymphocyte count above 200 cells/mm$^3$.

Exclusion criteria

- Patient suffering from active viral hepatitis or under treatment for hepatitis B or C.
- Patient suffering from a tumour that may require chemotherapy or radiotherapy.
- Patient suffering from hepatic (Child-Pugh score A, B, or C) or renal (estimated glomerular filtration rate <50 ml/min) failure.
- Patient suffering from a gastrointestinal pathology that may affect the absorption of dolutegravir (bypass, short hail, etc.).
- Patient suffering from thrombocytopenia, coagulation disorder, or medication that may cause bleeding (treatment with antiplatelet drugs or anticoagulants).
- Patient taking drug treatment responsible for significant interaction with dolutegravir according to the University of Liverpool website (https://www.hiv-druginteractions.org/).
- Pregnancy or breastfeeding.
- Any acute pathology within 8 weeks prior to inclusion.
- Any condition that, in the judgement of the investigators, could compromise the patient's adherence to the protocol.
- Patient already on dolutegravir twice daily or other integrase inhibitor therapy (elvitegravir or raltegravir).
- Patient with rectoanal pathology that contraindicates biopsy (e.g., anal stenosis).

# Appendix 2

## Comparisons of models for the virological markers

**Appendix 2—table 1.** Comparison of models for the virological markers (between-group analysis).

| Dependent variable | Explanatory variables | Treatment only (repeated-measures model) Estimate (95% CI) | p* | Treatment + time point (as discrete variable) Estimate (95% CI) | p | Treatment + time (as continuous variable) Estimate (95% CI) | p |
|---|---|---|---|---|---|---|---|
| Total DNA (log$_{10}$ fold change from baseline) | Treatment, intensified vs. control group | −0.38 (−0.55 to −0.21)† | 2.3 × 10$^{-5}$ | −0.38 (−0.55 to −0.20) | 5.8 × 10$^{-5}$ | −0.38 (−0.56 to −0.20) | 6.5 × 10$^{-5}$ |
| | Time, per day | - | - | - | 0.15 | 0.001 (−0.002 to 0.004) | 0.41 |
| US RNA (log$_{10}$ fold change from baseline) | Treatment, intensified vs. control group | −0.56 (−0.98 to −0.14) | 0.010 | −0.49 (−0.94 to −0.04) | 0.035 | −0.49 (−0.93 to −0.05) | 0.028 |
| | Time, per day | - | - | - | 0.99 | −0.001 (−0.008 to 0.006) | 0.82 |
| US RNA/total DNA ratio (log$_{10}$ fold change from baseline) | Treatment, intensified vs. control group | −0.36 (−0.77 to 0.06) | 0.090 | −0.29 (−0.73 to 0.16) | 0.20 | −0.28 (−0.71 to 0.16) | 0.21 |
| | Time, per day | - | - | - | 0.91 | −0.002 (−0.009 to 0.005) | 0.62 |

*p values were calculated by type III tests of fixed effects.
†Control group is assigned zero value.

**Appendix 2—table 2.** Comparison of models for the virological markers (within-group analysis).

| Dependent variable | Group | Explanatory variables | Intercept only (repeated-measures design) Estimate (95% CI) | p* | Intercept + time point Estimate (95% CI) | p |
|---|---|---|---|---|---|---|
| Total DNA (log$_{10}$ fold change from baseline) | Intensified group | Intercept | −0.21 (−0.33 to −0.08) | 0.0022 | −0.17 (−0.44 to 0.11) | 0.0016 |
| | | Time point | - | - | - | 0.40 |
| | Control group | Intercept | 0.16 (0.05 to 0.27) | 0.0053 | 0.24 (0.01 to 0.48) | 0.019 |
| | | Time point | - | - | - | 0.42 |
| US RNA (log$_{10}$ fold change from baseline) | Intensified group | Intercept | −0.54 (−0.75 to −0.33) | 6.0 × 10$^{-5}$ | −0.62 (−1.19 to −0.06) | 0.0069 |
| | | Time point | - | - | - | 0.60 |
| | Control group | Intercept | 0.05 (−0.26 to 0.37) | 0.73 | 0.24 (−0.44 to 0.92) | 0.71 |
| | | Time point | - | - | - | 0.88 |
| US RNA/total DNA ratio (log$_{10}$ fold change from baseline) | Intensified group | Intercept | −0.47 (−0.67 to −0.27) | 2.2 × 10$^{-4}$ | −0.53 (−1.09 to 0.03) | 0.029 |
| | | Time point | - | - | - | 0.59 |
| | Control group | Intercept | 0.01 (−0.29 to 0.32) | 0.94 | −0.07 (−0.76 to 0.62) | 0.81 |
| | | Time point | - | - | - | 0.86 |

*p values were calculated by type III tests of fixed effects.

## Appendix 3

## Gating strategy of the flow cytometry measurements

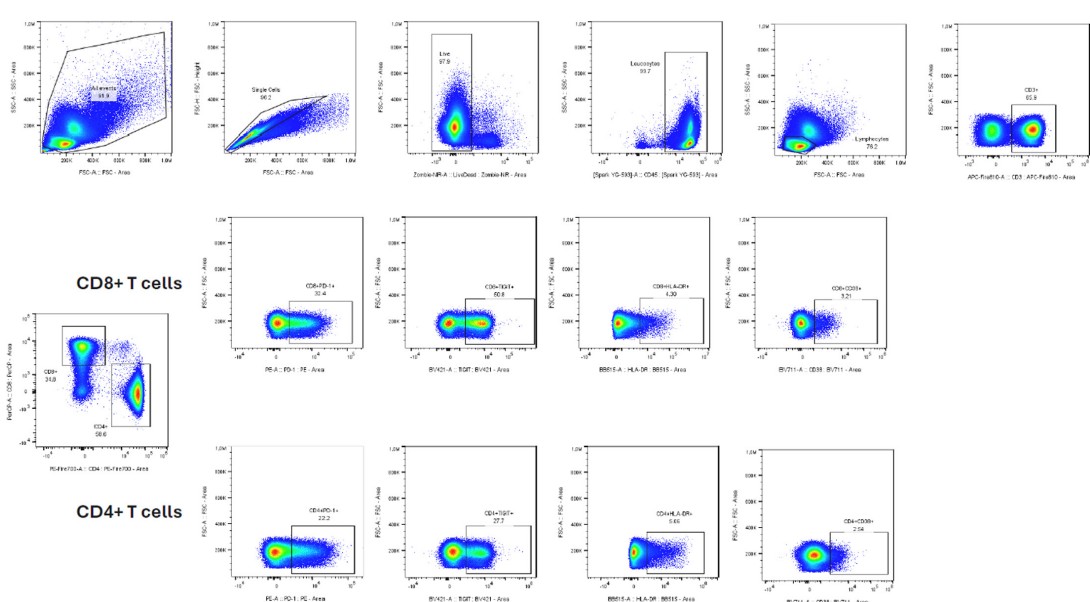

**Appendix 3—figure 1.** Gating strategy of the flow cytometry measurements.

