## [Editor Report · eLife Assessment]

This **valuable** clinical trial compares the impact of dolutegravir intensification on longitudinal measures of total HIV DNA and day 84 measures of intact HIV DNA. The trial was well-designed, and the paper is easy to read and provides hypothesis generation-level evidence that treatment intensification might decrease intact HIV DNA level in some people after 3 months. The findings are **solid**, with significant limitations being that study endpoints and hypotheses were not precisely defined prior to the trial, and that effect size is limited and inconsistent across trial participants.

---

## [Referee Report · Reviewer #1 (Public review)]

Fombellida-Lopez and colleagues describe the results of an ART intensification trial in people with HIV infection (PWH) on suppressive ART to determine the effect of increasing the dose of one ART drug, dolutegravir, on viral reservoirs, immune activation, exhaustion, and circulating inflammatory markers. The authors hypothesize that ART intensification will provide clues about the degree to which low-level viral replication is occurring in circulation and in tissues despite ongoing ART, which could be identified if reservoirs decrease and/or if immune biomarkers change. The trial design is straightforward and well-described, and the intervention appears to have been well tolerated. The investigators observed an increase in dolutegravir concentrations in circulation, and to a lesser degree in tissues, in the intervention group, indicating that the intervention has functioned as expected (ART has been intensified in vivo). Several outcome measures changed during the trial period in the intervention group, leading the investigators to conclude that their results provide strong evidence of ongoing replication on standard ART. The results of this small trial are intriguing, and a few observations in particular are hypothesis-generating and potentially justify further clinical trials to explore them in depth.

---

## [Referee Report · Reviewer #2 (Public review)]

Summary:

An intensification study with a double dose of 2nd generation integrase inhibitor with a background of nucleoside analog inhibitors of the HIV retrotranscriptase in 2, and inflammation is associated with the development of co-morbidities in 20 individuals randomized with controls, with an impact on the levels of viral reservoirs and inflammation markers. Viral reservoirs in HIV are the main impediment to an HIV cure, and inflammation is associated with co-morbidities.

Strengths:

The intervention that leads to a decrease of viral reservoirs and inflammation is quite straightforward forward as a doubling of the INSTI is used in some individuals with INSTI resistance, with good tolerability.

This is a very well documented study, both in blood and tissues, which is a great achievement due to the difficulty of body sampling in well-controlled individuals on antiretroviral therapy. The laboratory assays are performed by specialists in the field with state-of-the art quantification assays. Both the introduction and the discussion are remarkably well presented and documented.

The findings also have a potential impact on the management of chronic HIV infection.

---

## [Referee Report · Reviewer #3 (Public review)]

The introduction does a very good job of discussing the issue around whether there is ongoing replication in people with HIV on antiretroviral therapy. Sporadic, non-sustained replication likely occurs in many PWH on ART related to adherence, drug-drug interactions and possibly penetration of antivirals into sanctuary areas of replication and as the authors point out proving it does not occur is likely not possible and proving it does occur is likely very dependent on the population studied and the design of the intervention. Whether the consequences of this replication in the absence of evolution toward resistance have clinical significance challenging question to address.

It is important to note that INSTI-based therapy may have a different impact on HIV replication events that results in differences in virus release for specific cell type (those responsible for "second phase" decay) by blocking integration in cells that have completed reverse transcription prior to ART initiation but have yet to be fully activated. In a PI or NNRTI-based regimen, those cells will release virus, whereas with an INSTI-based regimen, they will not.

Given the very small sample size, there is a substantial risk of imbalance between the groups in important baseline measures.

Comments on the revised version from the editor:

I appreciate that the authors thoroughly address the reviewer's concerns in the response letter. Most importantly, they acknowledge that "The absence of a pre-specified statistical endpoint or sample size calculation reflects the exploratory nature of the trial." This is vital because the transient impact on total HIV DNA in the intensified versus standard dose arm raises questions about any sustained or meaningful anti-reservoir effect and was also not hypothesized a priori. The authors explanation that HIV DNA may have rebounded due to clonal expansion is interesting but not assessed directly in the trial.

The greater decrease in intact HIV DNA between days 0 and 84 in the intensified arm are notable but are somewhat limited by small sample size, small effect size and lack of data between these two timepoints.

Unfortunately, the hypothesis generating nature of the conclusions which is outlined nicely in the author's response letter is only acknowledged in the discussion of the revised paper. The abstract and results are only marginally different than the original version and still read as definitive when the evidence is only hypothesis generating. For these reasons, the level of evidence remains incomplete as before.

---

## [Author Response]

**Reviewer #1 (Public review):**
Fombellida-Lopez and colleagues describe the results of an ART intensification trial in people with HIV infection (PWH) on suppressive ART to determine the effect of increasing the dose of one ART drug, dolutegravir, on viral reservoirs, immune activation, exhaustion, and circulating inflammatory markers. The authors hypothesize that ART intensification will provide clues about the degree to which low-level viral replication is occurring in circulation and in tissues despite ongoing ART, which could be identified if reservoirs decrease and/or if immune biomarkers change. The trial design is straightforward and well-described, and the intervention appears to have been well tolerated. The investigators observed an increase in dolutegravir concentrations in circulation, and to a lesser degree in tissues, in the intervention group, indicating that the intervention has functioned as expected (ART has been intensified in vivo). Several outcome measures changed during the trial period in the intervention group, leading the investigators to conclude that their results provide strong evidence of ongoing replication on standard ART. The results of this small trial are intriguing, and a few observations in particular are hypothesis-generating and potentially justify further clinical trials to explore them in depth. However, I am concerned about over-interpretation of results that do not fully justify the authors' conclusions.

We thank Reviewer #1 for their thoughtful and constructive comments, which helped us clarify and improve the manuscript. Below, we address each of the reviewer’s points and describe the changes that we implemented in the revised version. We acknowledge the reviewer’s concern regarding potential overinterpretation of certain findings, and in the revised version we took particular care to ensure that all conclusions are supported by the data and framed within the exploratory nature of the study.

(1) Trial objectives: What was the primary objective of the trial? This is not clearly stated. The authors describe changes in some reservoir parameters and no changes in others. Which of these was the primary outcome? No a priori hypothesis / primary objective is stated, nor is there explicit justification (power calculations, prior in vivo evidence) for the small n, unblinded design, and lack of placebo control. In the abstract (line 36, "significant decreases in total HIV DNA") and conclusion (lines 244-246), the authors state that total proviral DNA decreased as a result of ART intensification. However, in Figures 2A and 2E (and in line 251), the authors indicate that total proviral DNA did not change. These statements are confusing and appear to be contradictory. Regarding the decrease in total proviral DNA, I believe the authors may mean that they observed transient decrease in total proviral DNA during the intensification period (day 28 in particular, Figure 2A), however this level increases at Day 56 and then returns to baseline at Day 84, which is the source of the negative observation. Stating that total proviral DNA decreased as a result of the intervention when it ultimately did not is misleading, unless the investigators intended the day 28 timepoint as a primary endpoint for reservoir reduction - if so, this is never stated, and it is unclear why the intervention would then be continued until day 84? If, instead, reservoir reduction at the end of the intervention was the primary endpoint (again, unstated by the authors), then it is not appropriate to state that the total proviral reservoir decreased significantly when it did not.

We agree with the reviewer that the primary objective of the study was not explicitly stated in the submitted manuscript. We clarified this in the revised manuscript (lines 361-364). As registered on ClinicalTrials.gov (NCT05351684), the primary outcome was defined as “To evaluate the impact of treatment intensification at the level of total and replication-competent reservoir (RCR) in blood and in tissues”, with a time frame of 3 months. Accordingly, our aim was to explore whether any measurable reduction in the HIV reservoir (total or replication-competent) occurred during the intensification period, including at day 28, 56, or 84. The protocol did not prespecify a single time point for this effect to occur, and the exploratory design allowed for detection of transient or sustained changes within the intensification window.

We recognize that this scope was not clearly articulated in the original text and may have led to confusion in interpreting the transient drop in total HIV DNA observed at day 28. While total DNA ultimately returned to baseline by the end of intensification, the presence of a transient reduction during this 3-month window still fits within the framework of the study’s registered objective. Moreover, although the change in total HIV DNA was transient, it aligns with the consistent direction of changes observed across the multiple independent measures, including CA HIV RNA, RNA/DNA ratio and intact HIV DNA, collectively supporting a biological effect of intensification.

We would also like to stress that this is the first clinical trial ever, in which an ART intensification is performed not by adding an extra drug but by increasing the dosage of an existing drug. Therefore, we were more interested in the overall, cumulative, effect of intensification throughout the entire trial period, than in differences between groups at individual time points. We clarified in the revised manuscript that this was a proof-of-concept phase 2 study, designed to reveal biological effects of ART intensification rather than confirm efficacy in a powered comparison. The absence of a prespecified statistical endpoint or sample size calculation reflects the exploratory nature of the trial.

(2) Intervention safety and tolerability: The results section lacks a specific heading for participant safety and tolerability of the intervention. I was wondering about clinically detectable viremia in the study. Were there any viral blips? Was the increased DTG well tolerated? This drug is known to cause myositis, headache, CPK elevation, hepatotoxicity, and headache. Were any of these observed? What is the authors' interpretation of the CD4:8 ratio change (line 198)? Is this a significant safety concern for a longer duration of intensification? Was there also a change in CD4% or only in absolute counts? Was there relative CD4 depletion observed in the rectal biopsy samples between days 0 and 84? Interestingly, T cells dropped at the same timepoints that reservoirs declined... how do the authors rule out that reservoir decline reflects transient T cell decline that is non-specific (not due to additional blockade of replication)?

We improved the Methods section to clarify how safety and tolerability were assessed during the study (lines 389-396). Safety evaluations were conducted on day 28 and day 84 and included a clinical examination and routine laboratory testing (liver function tests, kidney function, and complete blood count). Medication adherence was also monitored through pill counts performed by the study nurses.

No virological blips above 50 copies/mL were observed and no adverse events were reported by participants during the 3-month intensification period. Although CPK levels were not included in the routine biological monitoring, no participant reported muscle pain or other symptoms suggestive of muscle toxicity.

The CD4:CD8 ratio decrease noted during intensification was not associated with significant changes in absolute CD4 or CD8 counts, as shown in Figure 5. We interpret this ratio change as a transient redistribution rather than an immunological risk, therefore we do not consider it to represent a safety concern.

We would like to clarify that CD4⁺ T-cell counts did not significantly decrease in any of the treatment groups, as shown in Figure 5. The apparent decline observed concerns the CD4/CD8 ratio, which transiently dropped, but not the absolute number of CD4⁺ T cells. Moreover, although the dynamics of total HIV DNA is indeed similar to that of CD4/CD8 ratio (both declined transiently and then returned to baseline by day 84), the dynamics of unspliced RNA and unspliced RNA/total DNA ratio are clearly different, as these markers demonstrated a sustained decrease that was maintained throughout the trial period, even when the CD4/CD8 ratio already returned to baseline. Also, we observed a significant decrease in intact HIV DNA at day 84 compared to day 0. These effects cannot be easily explained by a transient decline in CD4+ cells.

(3) The investigators describe a decrease in intact proviral DNA after 84 days of ART intensification in circulating cells (Figure 2D), but no changes to total proviral DNA in blood or tissue (Figures 2A and 2E; IPDA does not appear to have been done on tissue samples). It is not clear why ART intensification would result in a selective decrease in intact proviruses and not in total proviruses if the source of these reservoir cells is due to ongoing replication. These reservoir results have multiple interpretations, including (but not limited to) the investigators' contention that this provides strong evidence of ongoing replication. However, ongoing replication results in the production of both intact and mutated/defective proviruses that both contribute to reservoir size (with defective proviruses vastly outnumbering intact proviruses). The small sample size and well-described heterogeneity of the HIV reservoir (with regard to overall size and composition) raise the possibility that the study was underpowered to detect differences over the 84-day intervention period. No power calculations or prior studies were described to justify the trial size or the duration of the intervention. Readers would benefit from a more nuanced discussion of reservoir changes observed here.

We sincerely thank the reviewer for this insightful comment. We fully agree that the reservoir dynamics observed in our study might raise several possible interpretations, and that its complexity, resulting from continuous cycles of expansion and contraction, reflects the heterogeneity of the latent reservoir.

Total HIV DNA in PBMCs showed a transient decline during intensification (notably at day 28), ultimately returning to baseline by day 84. This biphasic pattern likely reflects the combined effects of suppression of ongoing low-level replication by an increased DTG dosage, followed by the expansion of infected cell clones (mostly harbouring defective proviruses). In other words, the transient decrease in total (intact + defective) DNA at day 28 may be due to an initial decrease in newly infected cells upon ART intensification, however at the subsequent time points this effect was masked by proliferation (clonal expansion) of infected cells with defective proviruses. Recent studies suggest that intact and defective proviruses are subjected to different selection pressures by the immune system on ART (PMID: 38337034) and their decay on therapy is different (intact proviruses are cleared much more rapidly than defectives). In addition, defective proviruses can be preferentially expanded as they can reprogram the host cell proliferation machinery (https://doi.org/10.1101/2025.09.22.676989). This explains why in our study the intact proviruses decreased, but the total proviruses did not change, between days 0 and 84, in the intensification group. Interestingly, in the control group, we observed a significant increase in total DNA at day 84 compared to day 0, with no difference for the intact DNA, which is also in line with the clonal expansion of defective proviruses.

Importantly, we observed a significant decrease in intact proviral DNA between day 0 and day 84 in the intensification group (Figure 2D). This result directly addresses the study’s primary objective: assessing the impact of intensification on the replication-competent reservoir. In comparison, as the reviewer rightly points out, total HIV DNA includes over 90% defective genomes, which limits its interpretability as a biomarker of biologically relevant reservoir changes. In addition, other reservoir markers, such as cell-associated unspliced RNA and RNA/DNA ratios, also showed consistent trends supporting a biologically relevant effect of intensification. Even in the absence of sustained changes in total HIV DNA, the coherence across the different independent measures of the reservoir (intact DNA, unspliced RNA), suggests an effect indicative of ongoing replication pre-intensification.

Regarding tissue reservoirs, the lack of substantial change in total HIV DNA between days 0 and 84 is also in line with the predominance of defective sequences in these compartments. Moreover, the limited increase in rectal tissue dolutegravir levels during intensification (from 16.7% to 20% of plasma concentrations) may have limited the efficacy of the intervention in this site.

As for the IPDA on rectal biopsies, we attempted the assay using two independent DNA extraction methods (Promega Reliaprep and Qiagen Puregene), but both yielded high DNA shearing index values, and intact proviral detection was successful in only 3 of 40 samples. Given the poor DNA integrity, these results were not interpretable.

That said, we fully acknowledge the limitations of our study, especially the small sample size, and we agree with the reviewer that caution is needed when interpreting these findings. In the revised manuscript, we adopted a more measured tone in the discussion (lines 340-346), stating that these observations are exploratory and hypothesis-generating, and require confirmation in larger, more powered studies. Nonetheless, we believe that the convergence of multiple reservoir markers pointing in the same direction constitutes a meaningful biological effect that deserves further investigation.

(4) While a few statistically significant changes occurred in immune activation markers, it is not clear that these are biologically significant. Lines 175-186 and Figure 3: The change in CD4 cells + for TIGIT looks as though it declined by only 1-2%, and at day 84, the confidence interval appears to widen significantly at this timepoint, spanning an interquartile range of 4%. The only other immune activation/exhaustion marker change that reached statistical significance appears to be CD8 cells + for CD38 and HLA-DR, however, the decline appears to be a fraction of a percent, with the control group trending in the same direction. Despite marginal statistical significance, it is not clear there is any biological significance to these findings; Figure S6 supports the contention that there is no significant change in these parameters over time or between groups. With most markers showing no change and these two showing very small changes (and the latter moving in the same direction as the control group), these results do not justify the statement that intensifying DTG decreases immune activation and exhaustion (lines 38-40 in the abstract and elsewhere).

We agree with the reviewer that the observed changes in immune activation and exhaustion markers were modest. We revised the abstract and the manuscript text (including a section header) to reflect this more accurately (lines 39, 175, 185, 253). We noted that these differences, while statistically significant (e.g., in TIGIT+ CD4+ T cells and CD38+HLA-DR+ CD8+ T cells), were limited in magnitude. We explicitly acknowledged these limitations and interpreted the findings with appropriate caution.

(5) There are several limitations of the study design that deserve consideration beyond those discussed at line 327. The study was open-label and not placebo-controlled, which may have led to some medication adherence changes that confound results (authors describe one observation that may be evidence of this; lines 146-148). Randomized/blinded / cross-over design would be more robust and help determine signal from noise, given relatively small changes observed in the intervention arm.There does not seem to be a measurement of key outcome variables after treatment intensification ceased - evidence of an effect on replication through ART intensification would be enhanced by observing changes once intensification was stopped. Why was intensification maintained for 84 days? More information about the study duration would be helpful. Table 1 indicates that participants were 95% male. Sex is known to be a biological variable, particularly with regard to HIV reservoir size and chronic immune activation in PWH. Worldwide, 50% of PWH are women. Research into improving management/understanding of disease should reflect this, and equal participation should be sought in trials. Table 1 shows differing baseline reservoir sizes between the control and intervention groups. This may have important implications, particularly for outcomes where reservoir size is used as the denominator.

We expanded the limitations section to address several key aspects raised by the reviewer: the absence of blinding and placebo control, the predominantly male study population, and the lack of postintervention follow-up. While we acknowledge that open-label designs can introduce behavioural biases, including potential changes in adherence, we now explicitly state that placebo-controlled, blinded trials would provide a more robust assessment and are warranted in future research (lines 340346).

The 84-day duration of intensification was chosen based on previous studies and provided sufficient time for observing potential changes in viral transcription and reservoir dynamics. However, we agree that including post-intervention follow-up would have strengthened the conclusions, and we highlighted this limitation and future direction in the revised manuscript (lines 340-346).

The sex imbalance is now clearly acknowledged as a limitation in the revised manuscript, and we fully support ongoing efforts to promote equitable recruitment in HIV research. We would like to add that, in our study, rectal biopsies were coupled with anal cancer screening through HPV testing. This screening is specifically recommended for younger men who have sex with men (MSM), as outlined in the current EACS guidelines (see: https://eacs.sanfordguide.com/eacs part2/cancer/cancerscreening-methods). As a result, MSM participants had both a clinical incentive and medical interest to undergo this procedure, which likely contributed to the higher proportion of male participants in the study.

Lastly, although baseline total HIV DNA was higher in the intensified group, our statistical approach is based on a within-subject (repeated-measures) design, in which the longitudinal change of a parameter within the same participant during the study was the main outcome. In other words, we are not comparing absolute values of any marker between the groups, we are looking at changes of parameters from baseline within participants, and these are not expected to be affected by baseline imbalances.

(6) Figure 1: the increase in DTG levels is interesting - it is not uniform across participants. Several participants had lower levels of DTG at the end of the intervention. Though unlikely to be statistically significant, it would be interesting to evaluate if there is a correlation between change in DTG concentrations and virologic / reservoir / inflammatory parameters. A positive relationship between increasing DTG concentration and decreased cell-associated RNA, for example, would help support the hypothesis that ongoing replication is occurring.

We agree with the reviewer that assessing correlations between DTG concentrations and virological, immunological, or inflammatory markers would be highly informative. In fact, we initially explored this question in a preliminary way by examining whether individuals who showed a marked increase in DTG levels after intensification also demonstrated stronger changes in the viral reservoir. While this exploratory analysis did not reveal any clear associations, we would like to emphasize that correlating biological effects with DTG concentrations measured at a single timepoint may have limited interpretability. A more comprehensive understanding of the relationship between drug exposure and reservoir dynamics would ideally require multiple pharmacokinetic measurements over time, including pre-intensification baselines. This is particularly important given that DTG concentrations vary across individuals and over time, depending on adherence, metabolism, and other individual factors.

(7) Figure 2: IPDA in tissue- was this done? scRNA in blood (single copy assay) - would this be expected to correlate with usCaRNA? The most unambiguous result is the decrease in cell-associated RNA - accompanying results using single-copy assay in plasma would be helpful to bolster this result.

As mentioned in our response to point 3, we attempted IPDA on tissue samples, but technical limitations prevented reliable detection of intact proviruses. Regarding residual viremia, we did perform ultra-sensitive plasma HIV RNA quantification but due to a technical issue (an inadvertent PBMC contamination during plasma separation) that affected the reliability of the results we felt uncomfortable including these data in the manuscript.

The use of the US RNA / Total DNA ratio is not helpful/difficult to interpret since the control and intervention arms were unmatched for total DNA reservoir size at study entry.

We respectfully disagree with this comment. The US RNA/total DNA ratio is commonly used to assess the relative transcriptional activity of the viral reservoir, rather than its absolute size. While we acknowledge that the total HIV-1 DNA levels differed at baseline between the two groups, the US RNA/total DNA ratio specifically reflects the relationship between transcriptional activity and reservoir size within each individual, and is therefore not directly confounded by baseline differences in total DNA alone.

Moreover, our analyses focus on within-subject longitudinal changes from baseline, not on direct between-group comparisons of absolute marker values. As such, the observed changes in the US RNA/total DNA ratio over time are interpreted relative to each participant's baseline, mitigating concerns related to baseline imbalances between groups.

**Reviewer #2 (Public review):**
Summary:An intensification study with a double dose of 2nd generation integrase inhibitor with a background of nucleoside analog inhibitors of the HIV retrotranscriptase in 2, and inflammation is associated with the development of co-morbidities in 20 individuals randomized with controls, with an impact on the levels of viral reservoirs and inflammation markers. Viral reservoirs in HIV are the main impediment to an HIV cure, and inflammation is associated with co-morbidities.Strengths:The intervention that leads to a decrease of viral reservoirs and inflammation is quite straightforward forward as a doubling of the INSTI is used in some individuals with INSTI resistance, with good tolerability.This is a very well documented study, both in blood and tissues, which is a great achievement due to the difficulty of body sampling in well-controlled individuals on antiretroviral therapy. The laboratory assays are performed by specialists in the field with state-of-the art quantification assays. Both the introduction and the discussion are remarkably well presented and documented.The findings also have a potential impact on the management of chronic HIV infection.Weaknesses:I do not think that the size of the study can be considered a weakness, nor the fact that it is open-label either.

We thank Reviewer #2 for their constructive and supportive comments. We appreciate their positive assessment of the study design, the translational relevance of the intervention, and the technical quality of the assays. We also take note of their perspective regarding sample size and study design, which supports our positioning of this trial as an exploratory, hypothesis-generating phase 2 study.

**Reviewer #3 (Public review):**
The introduction does a very good job of discussing the issue around whether there is ongoing replication in people with HIV on antiretroviral therapy. Sporadic, non-sustained replication likely occurs in many PWH on ART related to adherence, drug-drug interactions and possibly penetration of antivirals into sanctuary areas of replication and as the authors point out proving it does not occur is likely not possible and proving it does occur is likely very dependent on the population studied and the design of the intervention. Whether the consequences of this replication in the absence of evolution toward resistance have clinical significance challenging question to address.It is important to note that INSTI-based therapy may have a different impact on HIV replication events that results in differences in virus release for specific cell type (those responsible for "second phase" decay) by blocking integration in cells that have completed reverse transcription prior to ART initiation but have yet to be fully activated. In a PI or NNRTI-based regimen, those cells will release virus, whereas with an INSTI-based regimen, they will not.Given the very small sample size, there is a substantial risk of imbalance between the groups in important baseline measures. Unfortunately, with the small sample size, a non-significant P value is not helpful when comparing baseline measures between groups. One suggestion would be to provide the full range as opposed to the inter-quartile range (essentially only 5 or 6 values). The authors could also report the proportion of participants with baseline HIV RNA target not detected in the two groups.

We thank Reviewer #3 for their thoughtful and balanced review. We are grateful for the recognition of the strength of the Introduction, the complexity of evaluating residual replication, and the technical execution of the assays. We also appreciate the insightful suggestions for improving the clarity and transparency of our results and discussion.

We revised the manuscript to address several of the reviewer’s key concerns. We agree that the small sample size increases the risk of baseline imbalances. We acknowledged these limitations in the manuscript (lines 327-330). For transparency, we now provide both the full range and the IQR for all parameters in Table 1. However, we would like to stress that our statistical approach is based on a within-subject (repeated-measures) design, in which the longitudinal change of a parameter within the same participant during the study was the main outcome. In other words, we are not comparing absolute values of any marker between the groups, we are looking at changes of parameters from baseline within participants, and these are not expected to be affected by baseline imbalances.

A suggestion that there is a critical imbalance between groups is that the control group has significantly lower total HIV DNA in PBMC, despite the small sample size. The control group also has numerically longer time of continuous suppression, lower unspliced RNA, and lower intact proviral DNA. These differences may have biased the ability to see changes in DNA and US RNA in the control group.

We acknowledge the significant baseline difference in total HIV DNA between groups, which we have clearly reported. However, the other variables mentioned, such as duration of continuous viral suppression, unspliced RNA levels, and intact proviral DNA, did not differ significantly between groups at baseline, despite differences in the median values (that are always present). These numerical differences do not necessarily indicate a critical imbalance.

Notably, there was no significant difference in the change in US RNA/DNA between groups (Figure 2C).

The nonsignificant difference in the change in US RNA/total DNA between groups is not unexpected, given the significant between-group differences for both US RNA and total DNA changes. Since the ratio combines both markers, it is likely to show attenuated between-group differences compared to the individual components. However, while the difference did not reach statistical significance (p = 0.09), we still observed a trend towards a greater reduction in the US RNA/total DNA ratio in the intervention group.

The fact that the median relative change appears very similar in Figure 2C, yet there is a substantial difference in P values, is also a comment on the limits of the current sample size.

Although we surely agree that in general, the limited sample size impacts statistical power, we would like to point out that in Figure 2C, while the medians may appear similar, the ranges do differ between groups. At days 56 and 84, the median fold changes from baseline are indeed close but the full interquartile range in the DTG group stays below 1, while in the control group, the interquartile range is wider and covers approximately equal distance above and below 1. This explains the difference in p values between the groups.

The text should report the median change in US RNA and US RNA/DNA when describing Figures 2A-2C.

These data are already reported in the Results section (lines 164–166): "By day 84, US RNA and US RNA/total DNA ratio had decreased from day 0 by medians (IQRs) of 5.1 (3.3–6.4) and 4.6 (3.1–5.3) fold, respectively (p = 0.016 for both markers)."

This statistical comparison of changes in IPDA results between groups should be reported. The presentation of the absolute values of all the comparisons in the supplemental figures is a strength of the manuscript.In the assessment of ART intensification on immune activation and exhaustion, the fact that none of the comparisons between randomized groups were significant should be noted and discussed.

We would like to point out that a statistically significant difference between the randomized groups was observed for the frequency of CD4⁺ T cells expressing TIGIT, as shown in Figure 3A and reported in the Results section (p = 0.048).

The changes in CD4:CD8 ratio and sCD14 levels appear counterintuitive to the hypothesis and are commented on in the discussion.Overall, the discussion highlights the significant changes in the intensified group, which are suggestive. There is limited discussion of the comparisons between groups where the results are less convincing.

We observed statistically significant differences between the randomized groups for total DNA (p<0.001) and US RNA (p=0.01), as well as for the frequency of CD4⁺ T cells expressing TIGIT (p=0.048). We would like to stress that US RNA is a key marker of residual replication as it is very sensitive to de novo infection events. As discussed in the manuscript (lines 291-294), a newly infected CD4+ T lymphocyte can contain hundreds to thousands of US HIV RNA copies at the peak of infection. Therefore, a change in the US RNA level upon ART intensification is a very sensitive indicator of new infections. The fact that for US RNA we observed both a significant reduction in the intensified group and a significant difference between the groups is a strong indicator that some new infections had been occurring prior to intensification.

The limitations of the study should be more clearly discussed. The small sample size raises the possibility of imbalance at baseline. The supplemental figures (S3-S5) are helpful in showing the differences between groups at baseline, and the variability of measurements is more apparent. The lack of blinding is also a weakness, though the PK assessments do help note 3TC levels rise substantially in both groups for most of the time on study (Figure S2).The many assays and comparisons are listed as a strength. The many comparisons raise the possibility of finding significance by chance. In addition, if there is an imbalance at baseline outcomes, measuring related parameters will move in the same direction.

We agree that the multiple comparisons raise the possibility of chance findings but would like to stress that in an exploratory study like this it is very important to avoid a type II error. In addition, the consistent directionality of the most relevant outcomes (US RNA and intact DNA) lends biological plausibility to the observed effects.

The limited impact on activation and inflammation should be addressed in the discussion, as they are highlighted as a potentially important consequence of intermittent, not sustained replication in the introduction.The study is provocative and well executed, with the limitations listed above. Pharmacokinetic analyses help mitigate the lack of blinding. The major impact of this work is if it leads to a much larger randomized, controlled, blinded study of a longer duration, as the authors point out.

Finally, we fully endorse the reviewer’s suggestion that the primary contribution of this study lies in its value as a proof-of-concept and foundation for future randomized, blinded trials of greater scale and duration. We highlighted this more clearly in the revised Discussion (lines 340-346).

**Reviewer #1 (Recommendations for the authors):**
(1) Lines 84-87: How would chronic immune activation/inflammation be expected to differ if viral antigen is being released from stable reservoirs rather than low-level replication?

This is a very insightful question. Although release of viral antigens from stable reservoirs could certainly also trigger immune activation/inflammation, the reservoir cells in PWH on long-term ART are constantly being negatively selected by the immune system (PMID: 38337034; PMID: 36596305) so that after a number of years on therapy, most proviruses are either transcriptionally silent or express only a low amount of viral RNA/antigen. Recent evidence suggests that these selected cells possess specific biological properties that include mechanisms that limit proviral gene expression (PMID: 36599977; PMID: 36599978). In comparison, low-level replication would result in de novo infection of unselected, activated CD4+ cells that are expected to produce much more viral antigen than preselected reservoir cells.

(2) Lines 249-253: There are multiple ways to explain this observation - alternatively, the total proviral DNA declined due to transient CD4 depletion.

As discussed above, CD4⁺ T-cell counts did not significantly decrease in any of the treatment groups, as shown in Figure 5. The apparent decline observed concerns the CD4/CD8 ratio, which transiently dropped, but not the absolute number of CD4⁺ T cells. Moreover, although the dynamics of total HIV DNA is indeed similar to that of CD4/CD8 ratio (both declined transiently and then returned to baseline by day 84), the dynamics of unspliced RNA and unspliced RNA/total DNA ratio is clearly different, as these markers demonstrated a sustained decrease that was maintained throughout the trial period. Also, we observed a significant decrease in intact HIV DNA at day 84 compared to day 0. These effects cannot be easily explained by a transient decline in CD4+ cells.

(3) Lines 301-305: This is a confusing explanation for not seeing an effect in tissue. Overall, there was no change in total proviral DNA in blood between days 0 and 84 either - yet the explanation for this observation is different (249-253). Was IPDA not performed on the tissue? Wouldn't this be the preferred test for reservoir depletion?

We thank the reviewer for bringing this point to our attention. We modified the Discussion to prevent the confusion (lines 303-305). As for the IPDA on tissue, we attempted this assay on the tissue samples using two independent DNA extraction methods (Promega Reliaprep and Qiagen Puregene), but both yielded high DNA shearing index values, and intact proviral detection was successful in only 3 of 40 samples. Given the poor DNA integrity, these results were not interpretable.